# Comprehensive proteomics and meta-analysis of COVID-19 host response

Haris Babačić [1] ✉, Wanda Christ [2], José Eduardo Araújo[1],
Georgios Mermelekas[1], Nidhi Sharma[1], Janne Tynell [2], Marina García [2],
Renata Varnaite[2], Hilmir Asgeirsson [3,4], Hedvig Glans[2,3], Janne Lehtiö [1],
Sara Gredmark-Russ[2,3,5], Jonas Klingström [2,6,7] & Maria Pernemalm[1,7] ✉

COVID-19 is characterised by systemic immunological perturbations in the human body, which can lead to multi-organ damage. Many of these processes are considered to be mediated by the blood. Therefore, to better understand the systemic host response to SARS-CoV-2 infection, we performed systematic analyses of the circulating, soluble proteins in the blood through global proteomics by mass-spectrometry (MS) proteomics. Here, we show that a large part of the soluble blood proteome is altered in COVID-19, among them elevated levels of interferon-induced and proteasomal proteins. Some proteins that have alternating levels in human cells after a SARS-CoV-2 infection in vitro and in different organs of COVID-19 patients are deregulated in the blood, suggesting shared infection-related changes. The availability of different public proteomic resources on soluble blood proteome alterations leaves uncertainty about the change of a given protein during COVID-19. Hence, we performed a systematic review and meta-analysis of MS global proteomics studies of soluble blood proteomes, including up to 1706 individuals (1039 COVID-19 patients), to provide concluding estimates for the alteration of 1517 soluble blood proteins in COVID-19. Finally, based on the meta-analysis we developed CoViMAPP, an open-access resource for effect sizes of alterations and diagnostic potential of soluble blood proteins in COVID-19, which is publicly available for the research, clinical, and academic community.

Abundance of evidence has demonstrated that the coronavirus disease 2019 (COVID-19), caused by the Severe Acute Respiratory Syndrome Coronavirus 2 (SARS-CoV-2), is a multisystemic disease[1]. The cytokine storm, i.e., an elevated release of cytokines in the blood that leads to systemic hyperinflammation, is considered one of the major patho-physiological mechanisms behind severe COVID-19, which leads to multi-organ damage, and can eventually cause death[2–4]. This makes the plasma, i.e., the liquid component of blood, a good biological material to explore systemic biological processes involved in COVID-19 patho-genesis and discover new biomarkers. Previously, different circulating molecules, such as lipids, metabolites, mRNAs, and proteins[5–10], have been investigated in COVID-19 patients. Still, to date, proteins remain the main circulating biomarkers for COVID-19 in clinical practice, aiding diagnosis and prognosis. Higher levels of well-established

[1]Science for Life Laboratory and Department of Oncology and Pathology, Karolinska Institutet, Stockholm, Sweden. [2]Centre for Infectious Medicine, Department of Medicine Huddinge, Karolinska Institutet, Stockholm, Sweden. [3]Department of Infectious Diseases, Karolinska University Hospital, Stockholm, Sweden. [4]Unit of Infectious Diseases, Department of Medicine Huddinge, Karolinska Institutet, Stockholm, Sweden. [5]The Laboratory for Molecular Infection Medicine Sweden (MIMS), Umeå, Sweden. [6]Division of Molecular Medicine and Virology (MMV), Department of Biomedical and Clinical Sciences (BKV), Linköping University, Linköping, Sweden. [7]These authors contributed equally: Jonas Klingström, Maria Pernemalm. ✉e-mail: haris.babacic@ki.se; maria.pernemalm@ki.se

biomarkers, such as the inflammatory proteins interleukin 6 (IL6) and C-reactive protein (CRP), the liver enzymes aspartate aminotransferase (AST) and alanine aminotransferase (ALT), and the fibrin degradation products (D-dimers), are among protein biomarkers for diagnosis and disease severity[11–13]. However, these proteins are widely used in practice and are not infection-specific clinical biomarkers. There is still a need to further identify biomarkers in relation to host response, organ involvement, and prognosis in COVID-19.

Several studies using affinity-based (AB) proteomic methods have quantified hundreds of proteins in plasma or serum of patients with COVID-19[9,14–23]. However, AB methods are targeted, focusing on proteins of interest, and depend on the specificity of the affinity molecule. In contrast, MS methods do not require an affinity molecule, but instead provide detection of a protein based on its amino-acid sequence, often performed as global proteomics. The two hitherto most in-depth studies of the soluble blood proteome (>1000 proteins) in COVID-19 were performed with AB methods, applying the Olink's antibody-based proximity extension assays (PEA)[16] and SOMAscan's aptamer platform[9]. Most studies using global MS methods in COVID-19 to date have identified several hundreds of proteins in plasma or serum[14], reporting varying estimates of alterations in blood levels. Given that the soluble blood proteome is reported to contain at least 4500 proteins[14], this still leaves a large portion of the proteome unexplored in COVID-19.

To expand the soluble blood proteome coverage, we here performed a comprehensive MS-based proteome profiling of serum samples in hospitalised COVID-19 patients by high-resolution isoelectric focusing (HiRIEF) coupled with liquid chromatography and mass-spectrometry (LC-MS/MS). We also performed in vitro SARS-CoV-2 infection experiments to compare to proteome and phosphoproteome changes occurring in the blood of COVID-19 patients. Due to the variable estimates of soluble blood proteome alterations in several previously published proteomics studies, by analysing 21 cohorts and up to 1706 individuals (1039 COVID-19 patients), we performed a meta-analysis of global soluble blood proteome alterations in COVID-19, to provide concluding estimates of alterations for 1517 soluble blood proteins and their potential to aid diagnosis (see Fig. 1).

In this work, we show that a large part of the soluble blood proteome is altered in COVID-19 patients. COVID-19 patients had elevated serum levels of NF-kB-, interferon-, purine metabolism-, heat shock-, and proteasomal- proteins, the latter of which had an association with anti-SARS-CoV-2 immune response and markers of severity. Dozens of these proteins also had a change in SARS-CoV-2-infected cells and tissues, such as the increase in the interferon-induced proteins ISG15, MX1, ISG20, and LAP3, and the proteasomal proteins PSMB5, PSMB7, PSMB8, PSMB10, and PSME1. Furthermore, we identified changes in phosphorylated peptides in the serum of COVID-19 patients and SARS-CoV-2-infected cells. Finally, we developed CoViMAPP, a comprehensive meta-analysis MS resource of soluble blood proteins' alterations in COVID-19.

## Results

### Patient characteristics
We analysed serum samples collected from 20 hospitalised patients with COVID-19 (15 men; age range: 34-67 years; median age: 53 years), infected with the ancestral SARS-CoV-2 variant, and 7 healthy controls (five men; age range: 26-53 years; median age: 31 years), which were PCR-negative and seronegative for SARS-CoV-2. Clinical and immune response details on the cohort are published elsewhere[24].

### Proteome coverage
HiRIEF LC-MS/MS (see Fig. 1a & 1c) provided in-depth proteome coverage, identifying 15,425 peptides mapping to 2037 proteins after gene-centric protein summarisation at 1% false discovery rate (FDR).

For comparison, currently the two most in-depth AB analyses of COVID-19, performed by PEA and SOMAscan aptamers, covered 1420 and 4563 proteins, respectively[9,16]. Of these, HiRIEF LC-MS/MS identified 530 proteins targeted by PEA and 1134 proteins targeted by SOMAscan (Fig. 2a). Some proteins were identified by all three methods, such as the higher-abundant proteins ITIH3, C2, C1QA, or the lower-abundant proteins CXCL16, CXCL12, NOTCH3, and ANXA11 (Fig. 2b). Still, 787 proteins were identified by HiRIEF LC-MS/MS that were not targeted by PEA and SOMAscan assays, including VTN, LRG1, the acute phase proteins orosomucoid 1 and 2 (ORM1 and ORM2), and the heat shock proteins HSPA4 and HSPA5 (Supplementary dataset 1). Compared to matched clinical chemistry assays' measurements in the same samples, HiRIEF LC-MS/MS measurements showed excellent agreement in quantifying levels of CRP and the liver enzymes AST and ALT (Fig. 2c), but slightly lesser agreement in lactate dehydrogenase (LDH) levels (Figure S1).

Taken together, these results show that HiRIEF LC-MS/MS provides in-depth profiles of the soluble blood proteome in COVID-19 patients, covers parts of the soluble blood proteome that have not been explored before, and maintains high precision compared to clinical assays.

### Differential comparison to PCR-negative controls
Principal component analysis (PCA) clearly separated COVID-19 samples from healthy controls (Fig. 3a), indicating large systemic perturbations in the serum proteome of hospitalised COVID-19 patients. This was further reflected in a differential analysis, where 619 of the analysed 1779 proteins had an alteration in protein serum levels in COVID-19 as compared to healthy controls (two-sided $t$ test, $p < 0.05$, 5% FDR, Fig. 3b, Supplementary Dataset 2). The interferon (IFN)-stimulated 15 KDa protein (ISG15) had the largest increase in serum of COVID-19 patients, followed by several other IFN-induced proteins, such as ILF2, MX1, ISG20, LAP3, and UBE2L6. Fifteen proteasomal proteins showed an increase in serum levels of COVID-19 patients, most notably PSMA7, PSME1, PSMB3, PSMA4 and PSMA5, whereas PSMD2 was the only one with decreased levels. In line with the cytokine storm hypothesis, several acute phase proteins were elevated in serum of COVID-19 patients, such as CRP, Lipopolysaccharide Binding Protein (LBP), ORM1, and ORM2. Hierarchical clustering of the differentially altered proteins (DAPs) without missing values ($n = 531$, 85.78% of all DAPs) identified two major protein clusters. One consisted of elevated serum proteins in COVID-19 involved in innate and adaptive immunity, cytokine signalling, post-translational modification, and neutrophil degranulation, whereas the other group consisted of proteins with decreased serum levels in COVID-19 that are involved in extracellular matrix organisation, haemostasis, and developmental biology (Fig. 3c). Among the proteins with the largest decrease were several proteins involved in fatty acid metabolism (such as LRP2 and apolipoproteins) and albumin, which can be due to the impaired nutrition often occurring during systemic inflammation.

### Comparison to PEA and SOMAscan
Comparing the DAPs identified in this study overlapping with those previously reported by AB methods ($p < 0.05$ and 5% FDR in both methods), HiRIEF LC-MS/MS showed very high categorical agreement to PEA or SOMAscan estimates (Fig. 3d, e, Supplementary datasets 3a–c). HiRIEF LC-MS/MS agreed in the direction of the alteration in 93% and 94% of the cases, compared to PEA and SOMAscan, respectively. However, comparing the statistical significance of the alterations showed higher agreement between HiRIEF LC-MS/MS and PEA than HiRIEF LC-MS/MS and SOMAscan. Among the 172 proteins overlapping in HiRIEF LC-MS/MS and PEA analyses, 102 were significant in both (59.30%); whereas among the

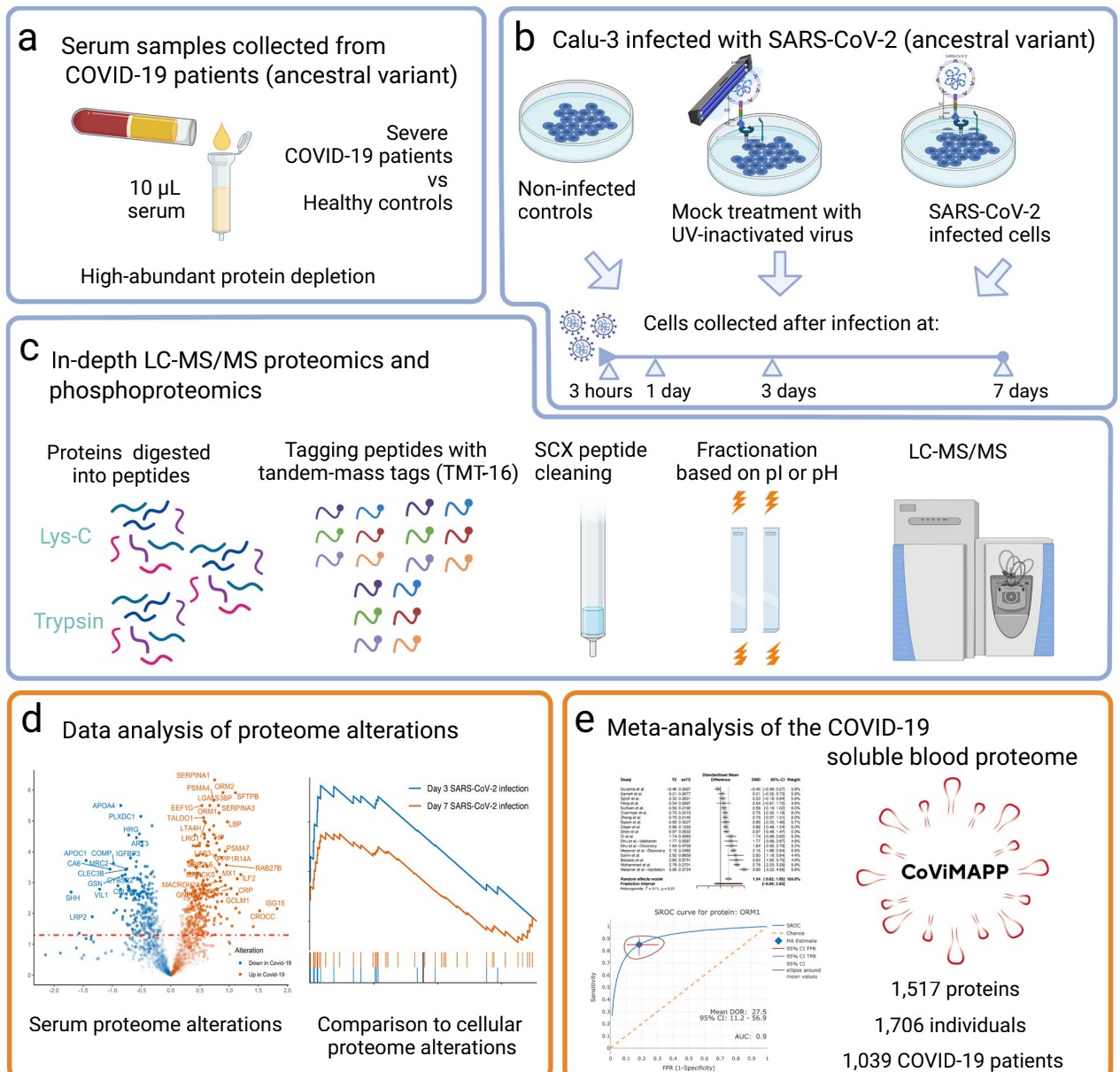

**Fig. 1 | Study workflow. a** The serum was depleted of 14 high-abundant proteins, to provide identification of lower abundant proteins; **b** We infected the Calu-3 line with SARS-CoV-2 in vitro and collected the cells at 3 hours, 1 day, 3 days, and 7 days after infection. Non-infected Calu-3 cells and Calu-3 cells treated with ultraviolet light-inactivated SARS-CoV-2 were also cultured as controls; **c** In-depth LC-MS/MS proteomics workflow; **d** Serum proteome alterations were traced to proteome alterations of SARS-CoV-2-infected cells; **e** Combining twenty global MS proteomics datasets with our dataset, we performed a per-protein meta-analysis of 1517 proteins identified in at least two cohorts and in up to 1706 individuals.

403 proteins overlapping in HiRIEF LC-MS/MS and SOMAscan analyses, only 103 were significant in both (25.56%). Still, we identified 443 proteins that were significant in our analysis that were not significant in either AB study, including elevated levels of fourteen proteasomal proteins, ILF2, CRP, SFTPB, ORM1, ORM2, UBE2L6, ISG20, and several heat shock proteins, i.e., isoforms A1 and B2 of HSP90A, HSPA4, HSPA8, and HSPA13. We detected a decrease in Angiotensin I Converting Enzyme (ACE), a paralogue of the SARS-CoV-2 cellular receptor ACE2. Importantly, HiRIEF LC-MS/MS detected alterations in 180 proteins that have not been targeted by either PEA or SOMAscan assays at all, including elevated levels of the brain-enriched proteins ZFHX3 and RELN, and decreased levels of the intestine-enriched VIL1 and NLRP6 (see Supplementary dataset 3a).

The agreement was similar when we adjusted the serum protein alterations in our analysis for age, sex, and comorbidities with limma[25]

models (Figure S2, Supplementary dataset 4) and lower compared to non-significant proteins in analyses with PEA and SOMAscan (Figure S3), likely due to technical bias and confounders.

In summary, we show that there is a large perturbation in the serum proteome of hospitalised COVID-19 patients. Using in-depth global MS proteomics, we describe soluble blood proteome alterations that have not been reported or targeted before with AB methods, such as proteasomal and heat shock proteins, and further validate several findings reported in two previous AB-based in-depth studies of the COVID-19 soluble blood proteome.

**Soluble blood proteins are traceable to SARS-CoV-2 infection**
To gain insights into which proteins derived from SARS-CoV-2 infection site, we performed an in vitro experiment infecting lung adenocarcinoma cultured human airway epithelial cells (Calu-3) with an

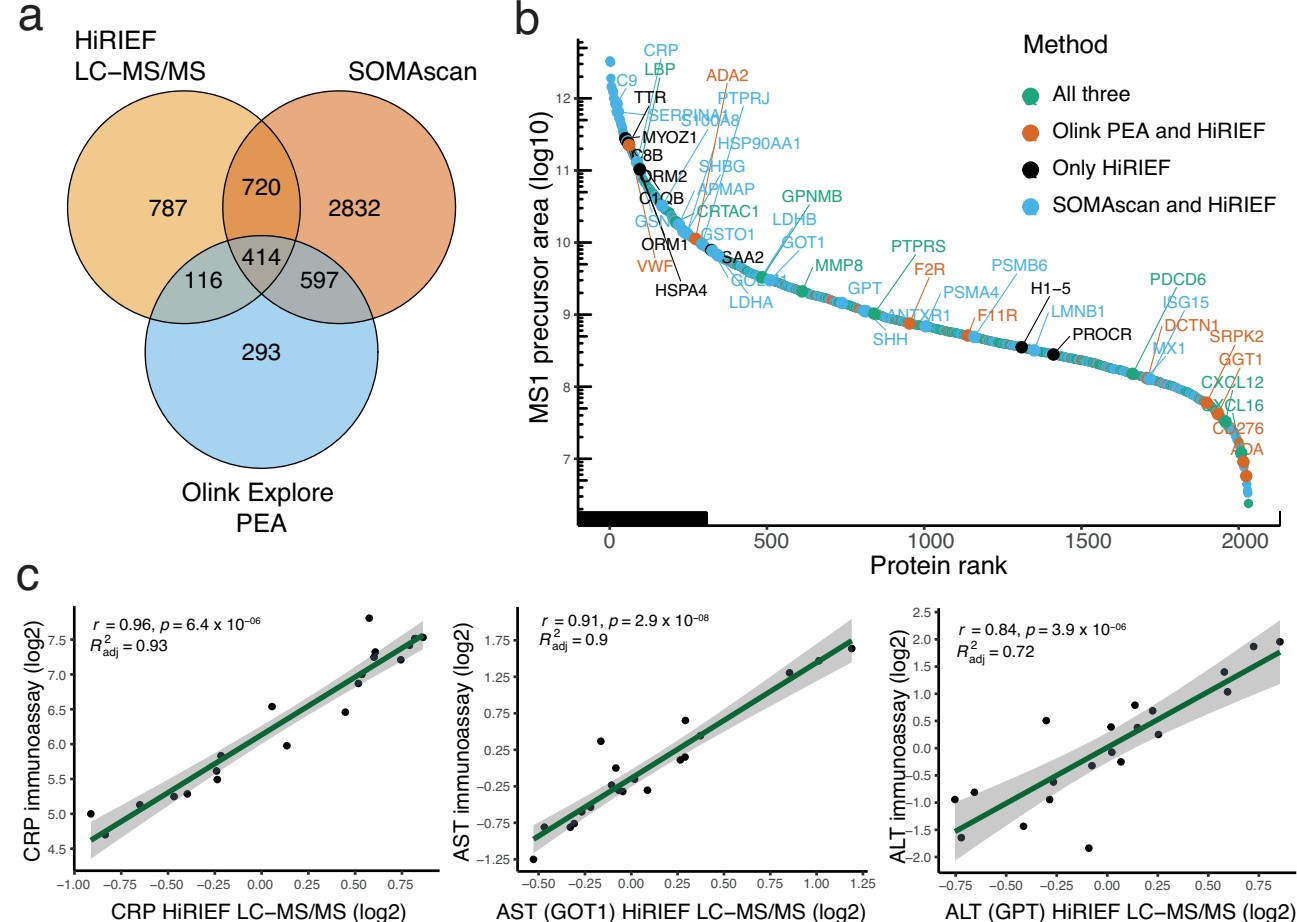

**Fig. 2 | Protein identification. a** Venn diagram of all identified proteins (gene-centric) in HiRIEF LC–MS/MS and overlap with all proteins identified and analysed with affinity-based proteomics[9,16]; **b** Proteins ranked (x axis) based on their median precursor area intensity (y axis), coloured according to the method of identification. **c** Agreement in quantifying the clinical biomarkers of COVID-19 severity between HiRIEF LC–MS/MS and clinical chemistry assays: CRP, AST (gene name: GOT1), and ALT (gene name: GPT). The line represents the linear regression fit line and the surrounding shaded area 95% confidence intervals (CI). The *p* values for the Spearman correlation coefficients (*r*) were obtained with a two-sided *t* test.

isolate of the ancestral variant (AV) of SARS-CoV-2. Samples were then collected 3 hours, 1 day, 3 days, and 7 days after infection (see Fig. 1b). In parallel, as controls, we collected samples on the same days from non-infected cells and from cells treated with SARS-CoV-2 virus inactivated by ultraviolet (UV) radiation. All conditions were performed in biological triplicates, and we performed proteomics analysis with high-pH fractionation and LC–MS/MS. We identified and quantified 10,336 human proteins and 62 sequence matches to SARS-CoV-2 proteins, with protein-centric summarisation of peptides. Thirteen viral protein sequences were quantified in all samples, with substantially higher levels in the infected cells at particularly day 3 and to a lesser extent at day 7 after infection (Figure S4). Six matched to the viral nucleocapsid phosphoprotein, five to the spike glycoprotein, and two to the membrane glycoprotein. Gene-centric summarisation of peptides identified 10,677 gene-matched proteins, of which 7875 with quantifications in all samples; this gene-centric protein matrix without missing values was used for further analyses.

We found no DAPs 3 hours and 1 day after infection ($p < 0.05$, 5% FDR, Figures S5a-b). However, a total of 532 and 893 proteins were DAPs in infected compared to non-infected cells, at 3 and 7 days after infection, respectively (two-sided *t* test, $p < 0.05$, 5% FDR, Figures S5c-d, Supplementary datasets 5a-b). Among the proteins with the largest increase in infected cells at 3 and 7 days after infection were several proteins involved in IFN-α and IFN-γ responses, such as the IFN-induced proteins IFIT1, IFIT2, IFIT3, and

IFIT5, the IFN-stimulated proteins ISG15 and ISG20, the viral RNases OAS1 and OAS2, and the antigen-presenting molecules class I – β-2 microglobulin (B2M) and HLA-A/-B/-E. Of the 532 DAPs deregulated at day 3, 44 proteins were also detected as altered in the serum of COVID-19 patients. Similarly, 77 of the 893 DAPs at day 7 were among DAPs in serum. Whereas more overlapping DAPs at day 3 were altered in serum of COVID-19 patients in the same direction (Fig. 4a, Supplementary dataset 5a), less than half overlapping DAPs at day 7 were altered in serum in the same direction (Fig. 4b, Supplementary dataset 5b). Still, 16 proteins had a consistent deregulation at both days of infection in the cell lines and in serum of COVID-19 patients at 5% FDR. Fifteen of these were upregulated, including the IFN-activated proteins ISG15, MX1, UBE2L6, ISG20, STAT1, the LAP3 peptidase, the proteasomal proteins PSME1, PSMB10 and PSMB8, TYMP, LGALS3BP, SERPINB1, WARS1, NAMPT, and THOP1. Only one protein—ALDOC—remained consistently downregulated across all comparisons. This set of proteins also had a statistically significant alteration in levels in the infected Calu-3 cells when compared to the Calu-3 cells receiving mock treatment with UV-inactivated SARS-CoV-2 (Fig. 4c, Figure S6). Sullivan et al. have previously reported elevated levels of ISG15, MX1, STAT1, and LGALS3BP in serum of COVID-19 patients by SOMAscan, but found no changes in UBE2L6, LAP3, TYMP, SERPINB1, NAMPT, or ALDOC[9]. Filbin et al. observed changes in LAP3, NAMPT, and THOP1 in plasma of COVID-19 patients, like our findings, but found no changes in

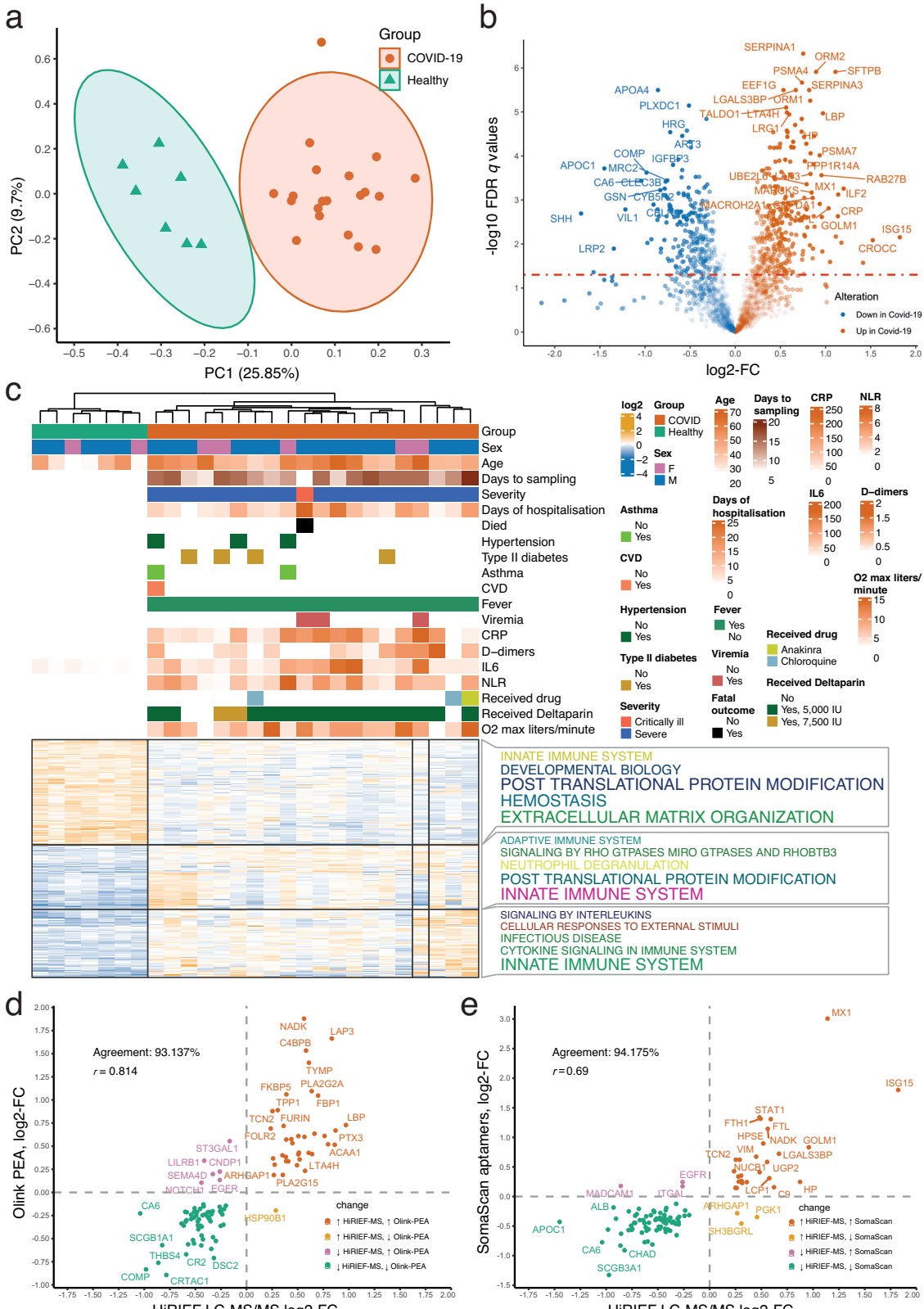

**Fig. 3 | COVID-19 serum proteomics by HiRIEF LC-MS/MS. a** PCA based on proteins without missing values; **b** Differential alteration of serum protein levels in COVID-19. The shade of the points represent the log2 fold change (log2-FC) multiplied by -log10 (adjusted *p* value); **c** Heatmap of differentially altered serum proteins and relation with clinical parameters. The five most frequent REACTOME terms describing a protein function per cluster are annotated; **d** Agreement with PEA profiling of the COVID-19 soluble blood proteome—overlapping proteins that are significant at 5% FDR in both methods. The agreement is represented as proportion

(in %) of proteins changing in the same direction out of the total number of overlapping proteins and with Spearman's correlation coefficient (*r*). The PEA analysis is adjusted for age, sex, ethnicity, heart disease, diabetes, hypertension, hyperlipidaemia, pulmonary disease, kidney disease, and immuno-compromised status as covariates; **e** Agreement with SOMAscan profiling of the COVID-19 serum proteome—overlapping proteins that are significant at 5% FDR in both methods. The SOMAscan analysis is adjusted for age and sex as covariates.

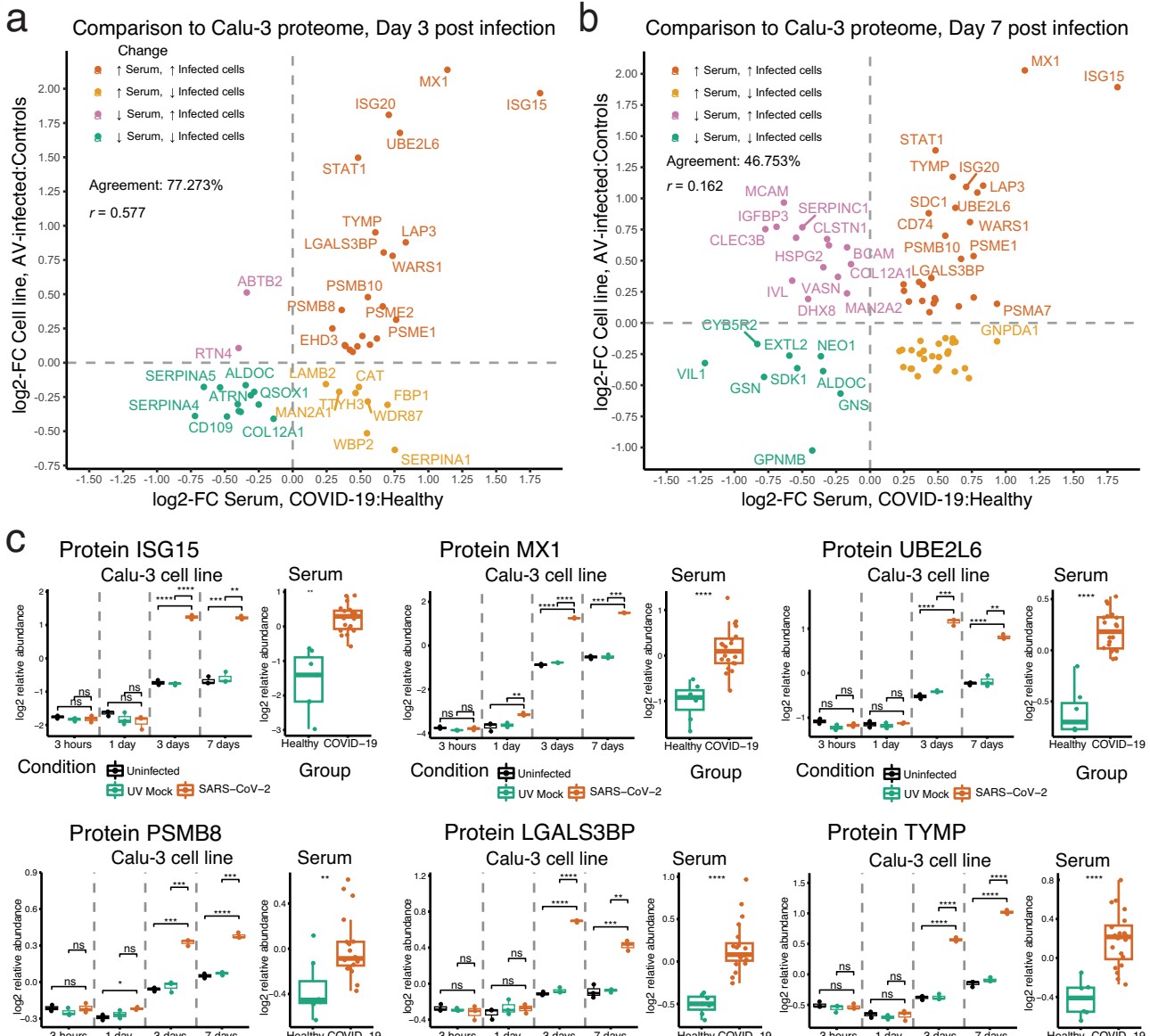

**Fig. 4 | Comparison of serum protein alterations to proteome alterations in Calu-3 cells infected with SARS-CoV-2. a** Day 3 after infection. The agreement is represented as proportion (in %) of proteins changing in the same direction out of the total number of overlapping proteins and with Spearman's correlation coefficient ($r$); **b** Day 7 after infection; **c** Boxplots for selected proteins consistently upregulated at day 3 and day 7 after infection, and in serum. The boxplots show protein levels in SARS-CoV-2-infected Calu-3 cells at different time points, compared to non-infected cells and cells treated with UV-inactivated SARS-CoV-2. All cells in each condition were cultured as biological replicates ($n = 3$ each). In addition, boxplots of serum levels of the respective protein in COVID-19 patients compared to healthy controls are presented. The box centre represents the median, the lower and upper box limits the 25th and 75th percentile, respectively, and whiskers' limits the minimum and maximum values of the data after removing outliers. ns = non-significant, * = $p < 0.05$, ** = $p < 0.01$, *** = $p < 0.005$, **** = $p < 0.001$. The $p$ values were determined with a two-sided $t$ test and adjusted for multiple testing with the FDR.

SERPINB1[16]. HiRIEF LC-MS/MS detected alterations in the remaining proteins, i.e., ISG20, PSMB8, PSMB10, ALDOC, and SERPINB1, emphasising the method's ability to capture alterations deriving from infection that are not picked up by other methods.

Altogether, we show that a portion of the altered serum proteins detected in COVID-19 patients showed the same alteration as in SARS-CoV-2-infected cells. Sets of proteins that were deregulated in the proteome of SARS-CoV-2-infected cells in the same direction (up or down) appeared also deregulated in the serum of COVID-19 patients, which suggests that alterations of these biological processes at the infection site can be traced in serum. Our in vitro data indicate that PSME1, PSMB10, PSMA7, and PSMB8 have likely been released from infected cells in the blood, which has not been reported before.

## Soluble blood proteins tissue origin

One distinction of plasma as a bodily fluid is that it has contact with most organs in the body. While this makes it informative about the body state, it also opens a question about the origin of the altered soluble blood proteins. It is established that some proteins, such as the acute phase proteins CRP, ORM1, ORM2, or the coagulation factors F5, F10, F11, are likely secreted by the liver during systemic inflammation and hence a general host response not specific to SARS-CoV-2 infection. However, for most proteins it is difficult to claim cellular or process-related specificity.

To trace the tissue origin of the serum proteins altered in COVID-19, we used tissue-enriched gene lists from the Human Protein Atlas (HPA) for annotation. Approximately 15% of the 1779 analysed proteins ($n = 265$) were tissue-enriched, comparable to estimates of the HPA,

where 15.5% of the protein-coding genes show enrichment in a particular tissue based on mRNA expression[26]. Similarly, 89 of the 619 altered soluble blood proteins (14.38%, Supplementary dataset 3a) were tissue -enriched, of which almost two thirds derive from the liver ($n$ = 57). Four of the five intestine-enriched proteins, i.e., ZG16, NLRP6, APOA4, and VIL1, showed decreased levels in serum of COVID-19 patients, whereas only CDHR2, a microvillar protein, showed elevated levels. Because LBP has often been considered as an inflammatory marker of gut leakage[27], we explored whether it correlated to intestinal markers. Although LBP correlated highly to other acute phase proteins, it surprisingly had a moderate negative correlation with all the intestinal markers except for CDHR2, with highest inverse correlation with the inflammasome protein NLRP6 (Spearman's $r$ = -0.742, $p$ = 0.000017, Supplementary Dataset 6), questioning whether this marker should be considered a gut leakage marker or a generic acute phase marker. All three proteins enriched in lymphoid organs, i.e., STAB2, Leukocyte Immunoglobulin-Like Receptor B1 (LILRB1), and CR2, had decreased serum levels in COVID-19 patients. LILRB1 is a receptor for class I MHC molecules that downregulates the immune response by inhibiting the FCER1A signalling[28,29], which in this study had a moderate negative correlation with acute phase proteins, such as CRP, LBP, ORM1, and ORM2. Hence, the lack of inhibition due to lower LILRB1 levels in patients with severe COVID-19 could contribute to hyperinflammation. Likely due to severe lung damage in our cohort of patients, we observed an increase in the lung-enriched surfactant B protein (SFTPB), expressed in type 2 alveolar cells, responsible for maintaining inflation of the alveoli.

Although some of the altered circulating proteins detected in our study are annotated as enriched in healthy tissues, a question remains whether they are altered in infected or damaged organs during COVID-19. To trace the altered serum proteins and identify which deregulated proteins during infection in our study are matching deregulated proteins in the organs during COVID-19, we intersected our findings with two datasets containing lists of DAPs in organs and tissues derived from patients who died of COVID-19, as compared to matched organ/tissue controls[30,31]. After filtering out well-annotated plasma proteins and immunoglobulins[30], many proteins identified in the different proteomics experiments in our study showed alteration in organs of COVID-19 patients, most of which were in the same direction (Figures S7-S9, Supplementary Dataset 7). Several proteins had a varying direction of the alteration in different organs as compared to the serum alterations, which could be due to different confounders. Still, 110 proteins showed consistent deregulation in the different organs compared to our findings (Figure S10, Supplementary Dataset 8). Eleven of the core set of 15 proteins that had consistently elevated levels in serum, and days 3 and 7 after infection in vitro, also had elevated levels in at least one human organ in vivo: NAMPT, ISG15, MX1, STAT1, TYMP, LAP3, ISG20, UBE2L6, the proteasomal proteins PSMB8 and PSMB10, and LGALS3BP. Additional 70 proteins upregulated at both days 3 and 7 after infection in vitro, were upregulated in at least one organ in vivo, including DDX58, TAP2, OAS2, OAS3, PARP9, IFIT1, IFIT2, IFIT3, IFI16, STAT3, MX2, and others (see Supplementary Dataset 7). Since most of these proteins showed no organ specificity, they are likely driving the systemic host response towards SARS-CoV-2, of which 11 proteins were DAPs in serum.

### Enrichment analyses and association with clinical parameters

To test whether COVID-19 organ-associated protein signatures can be identified in serum by HiRIEF LC-MS/MS, we performed gene-set enrichment analysis (GSEA)[32] on the serum proteins' alterations. For that purpose we filtered the organs' protein lists for previously annotated plasma proteins and immunoglobulins[30] and categorised all the proteins that were upregulated and downregulated in a specific organ of COVID-19 patients[30,31] as that organ's UP and DOWN protein set, respectively. We tested 33 protein sets and identified elevated serum levels of proteins belonging to upregulated protein sets in the lungs, lymph nodes, blood vessels, liver, and heart, and a down-regulated protein set in the spleen of deceased COVID-19 patients (permutation test, $p < 0.05$, 5% FDR, Fig. 5a, Supplementary dataset 9a). However, some of the proteins were shared between the organ sets, which could have increased the likelihood of enrichment due to shared systemic proteome alterations. Thus, to identify organ-specific changes, after filtering out proteins that were shared between protein sets—which could be systemic alterations occurring in SARS-CoV-2 infection—we could only detect a signal of elevated serum levels in COVID-19 patients of proteins downregulated in the white pulp of the spleen (permutation test, $p < 0.05$, 5% FDR, Fig. 5b, Supplementary dataset S9b), suggesting that the protein loss in the white pulp of the spleen occurring during COVID-19 might be due to the protein release in the bloodstream to fight infection.

To further unravel the biological processes behind the alterations, we performed GSEA on the gene sets curated in the Molecular Signature Database (MSigDb)[33]. GSEA of the MSigDb hallmark gene sets showed strong enrichment for the IFN-α and IFN-γ response, in line with our initial observations of high elevated levels of IFN-activated proteins in both the cell lines and serum of COVID-19 patients (Fig. 5c, Supplementary dataset 9c). Furthermore, GSEA of both the KEGG and REACTOME pathways confirmed our observations that the proteasome pathway is enriched in the serum proteome of COVID-19 patients (Fig. 5d, Supplementary datasets 9d-e). The FCERI Mediated NF-kB activation had the highest enrichment score (Figure S11)[34]. The GSEA of REACTOME pathways showed that proteasomal proteins, which we identified as elevated in the serum of COVID-19 patients and in SARS-CoV-2-infected cells, are also upregulated during HIV infection (see Supplementary dataset 9e), suggesting that this process is likely a shared antiviral response. Our results demonstrate the power of HiR-IEF LC-MS/MS as the only MS method that tracked these cellular alterations of host response at a systemic level in the blood.

We have previously characterised the anti-SARS-CoV-2 immune response of our COVID-19 cohort[24], which allowed us to explore how protein serum levels in the blood quantified by HiRIEF LC-MS/MS correlated to clinical parameters and immune response. A subset of proteins, among them VIL1, had negative correlation with anti-SARS-CoV-2 immune response and positive correlation with clinical markers of severity, such as hospitalisation duration, IL6, CRP, procalcitonin levels, or neutrophil-to-lymphocyte ratio (Figure S12 & S13, Supplementary Dataset 10). Another subset of proteins had a moderate to high positive correlation with activation of the adaptive immune system, i.e., higher anti-SARS-CoV-2 IgM, IgA, IgG levels, and higher percentage of activated CD8+ cells, while being at the same time associated with higher levels of ferritin, AST, ALT, and LDH, all of which have been associated with COVID-19 severity[13]. Within this subset were most of the proteasomal proteins, of which almost all had a moderate to high positive correlation with ferritin levels, AST, and ALT, while having moderate positive association with anti-SARS-CoV-2 IgM, IgA, and IgG levels (Figure S13). This suggests that while proteasomal proteins are involved in activating the immune response in COVID-19, they may also contribute to organ damage by hyperinflammation.

### Phosphoproteomics

To identify phosphorylated peptides in the serum we researched the data to include phosphorylation as a protein post-translational modification (PTM). We identified 865 phosphorylated peptides at 1% FDR, mapping to 233 proteins. Comparing the serum levels of 306 peptides with more confident phosphosite identifications (with false localisation rate (FLR) < 5%) showed alteration in 74 phosphorylated peptides in COVID-19 patients (two-sided $t$ test, $p < 0.05$, 20% FDR, Figure S14, Supplementary Dataset 11a). These included elevated levels of phosphorylated LGALS3BP (Y442-p), SERPINA3 (S384-p), ORM1 (S143-p),

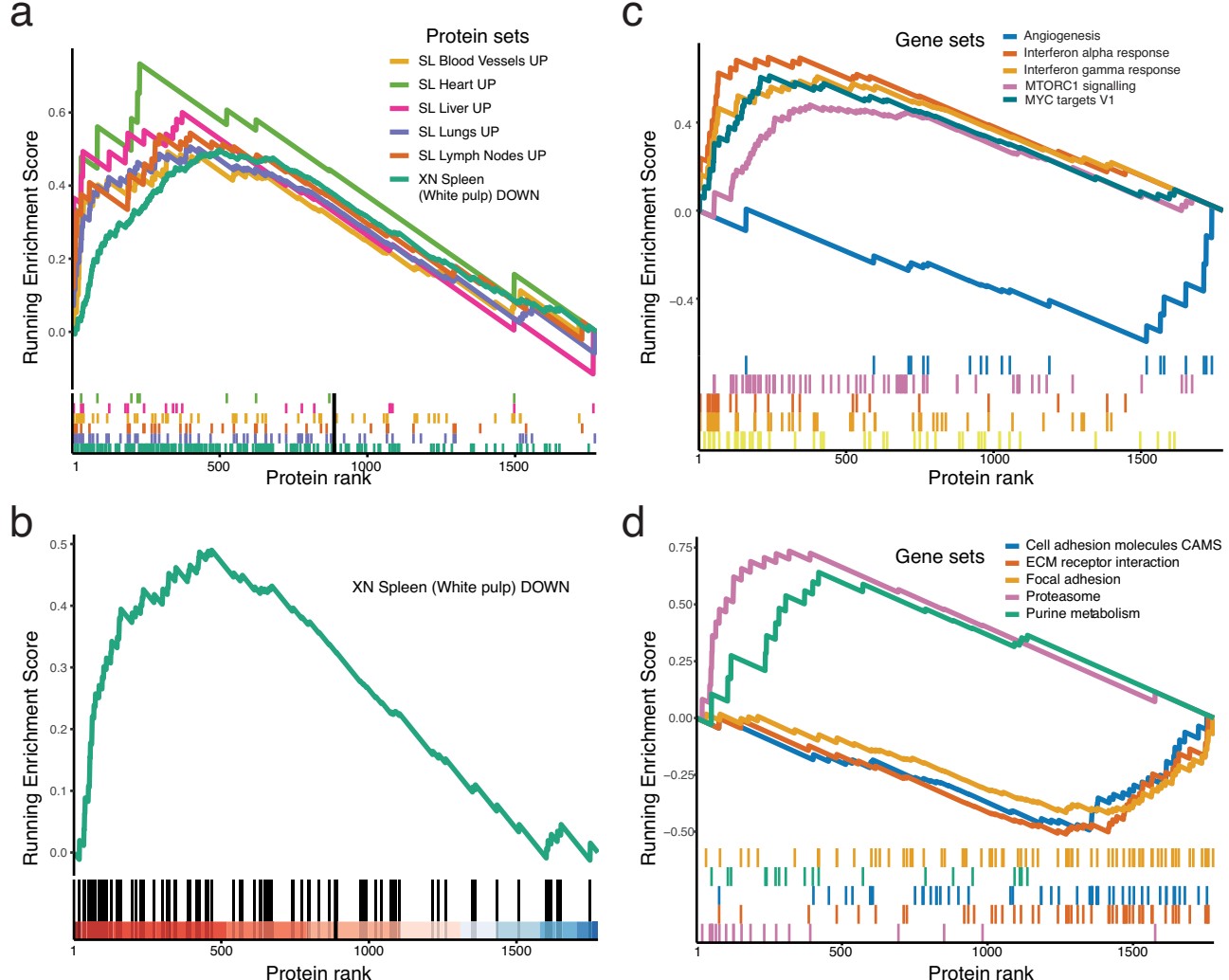

**Fig. 5 | Gene set enrichment analyses of organ-associated protein sets and MsigDb gene sets in serum of COVID-19 patients. a** Enriched organ-associated protein sets at 5% FDR, permutation test. The sets consist of proteins deregulated in a specific organ obtained from COVID-19 patients, regardless if the protein was identified in another organ-associated protein set. We tested 33 protein sets (17 upregulated and 16 downregulated) from 10 organs: adrenal glands, blood vessels, brain, heart, kidney, liver, lungs, lymph nodes, spleen (white pulp), and thyroid; **b** Enriched organ-specific protein set that was downregulated in the white pulp of the spleen, filtered for any protein that was downregulated in another organ in the same dataset[31]; **c** Enriched hallmark gene sets at 5% FDR; **d** Enriched KEGG pathways at 5% FDR. The rank (x axis) of the protein belonging in the gene set is based on log2-FC comparing mean serum levels between COVID-19 and PCR-negative healthy controls. The enrichment score is plotted on the y axis.

and C3 (T582-p), and the downregulated FN1 (S2432-p), CLEC3B (T159-p), ITIH1 (ENSP00000395836.1 S181-p), and GSN (T201-p).

To trace whether the identified phosphorylated peptides in serum occur during SARS-CoV-2 infection, we utilised a previously established protocol, to enrich the material from the in vitro experiment for phosphogroups and performed LC-MS/MS. We identified 18,127 phosphorylated peptides at 1% FDR, mapping to 3973 proteins, of which 15,022 peptides had FLR < 5%. Of the latter, 3407 were observed in all samples and statistically tested. Again, we observed no changes 3 hours and 1 day after infection at 5% FDR. Three days after infection, the largest increase was observed in phosphosites of the interferon-induced IFIH1 (S301-p) and STAT1 (S727-p), the transcription factor SP (ENSP00000375902.3 S378-p and S244-p), and a protein involved in trafficking of amino acids - CLTRN (S177-p) (two-sided $t$ test, $p < 0.05$, 5% FDR, Figure S15a, Supplementary Dataset 11b), whereas only the heat shock protein HSP90AB1 (ENSP00000325875.3 S226-p) had an increase 7 days after infection (two-sided $t$ test, $p < 0.05$, 5% FDR, Supplementary Dataset 11c). Regardless of the alteration in the

infected cells, 16 phosphorylated peptides identified in the serum had overlapping sequences with 30 phosphorylated peptides identified in the infected cells. Of these, 13 phosphorylated peptides were on the same residue, mapping to the proteins CANX, SPP1, AHSG, TGFB1I1, NAALADL2, GOLM1, and CALU. Only one of these phosphorylated peptides, SDAEEDGGTVsQEEEDRKPK, which was mapping to calnexin (CANX) S564-p, had decreased levels in the serum of COVID-19 patients (log2-FC = -0.227, $p = 0.005$, FDR $q = 0.049$, Figure S16). Interestingly, the levels of the canonical calnexin protein had no change in the serum of our group of COVID-19 patients, but it was upregulated in the SARS-CoV-2-infected cells 7 days after infection, and in lungs, white pulp of the spleen, and kidneys of patients who died of COVID-19[30,31]. In contrast, a phosphorylated peptide mapping to calnexin (AEEDEILNRsPR, CANX S583-p) had decreased levels in lung tissue of COVID-19 patients, as compared to healthy lung tissue[30]. These findings suggest that the phosphorylated calnexin levels are lower in vivo in COVID-19 patients, in contrast to the higher levels of non-phosphorylated calnexin, with potentially different roles in SARS-CoV-2 infection.

## Systematic literature review

The availability of MS-based global soluble blood proteomics datasets provided an opportunity to perform a meta-analysis on global soluble blood proteome alterations in COVID-19. Meta-analytical estimates provide the highest level of evidence, clarifying the direction and the statistical significance of alterations in instances where the results are conflicting. To the best of our knowledge, this is the first meta-analysis of its sort analysing global soluble blood proteome alterations in COVID-19.

To identify MS global proteomics studies relevant for the research question, we performed a systematic review of the literature and identified 18 studies (Figure S17, Supplementary dataset 12) that met the inclusion criteria[9,10,35–50]. Combining the published datasets with our dataset, we analysed 3475 soluble blood proteins across 21 proteome-profiling cohorts, including 1706 participants, of whom 1039 COVID-19 patients and 667 PCR-negative controls. Details on the studies are available in CoViMAPP ([https://doi.org/10.17044/scilifelab.22293148](https://doi.org/10.17044/scilifelab.22293148)). There was an inverse association between the number of

analysed samples and the number of identified proteins (Fig. 6a). HiRIEF LC-MS/MS was the only method that combined depletion, fractionation, and TMT-16 labelling, which favoured proteome coverage over throughput and had the highest number of proteins that could be included in the meta-analysis (Fig. 6b). In total, 1517 soluble blood proteins (43.65%) were identified in at least two studies, in at least three COVID-19 patients, and at least three SARS-CoV-2 PCR-negative controls per study. These proteins were included in the meta-analyses. HiRIEF LC-MS/MS's depth contributed to expanding the meta-analysis coverage, by quantifying 1239 of the 1517 proteins included in the meta-analysis (81.67%, Fig. 6c, d).

## Meta-analysis of soluble blood proteome alterations in COVID-19

To provide meta-analytical summary estimates of soluble blood proteome alterations in COVID-19, we summarised abundance estimates into standardised mean differences (SMD) between COVID-19 patients and PCR-negative controls of log2-normalised levels of soluble blood

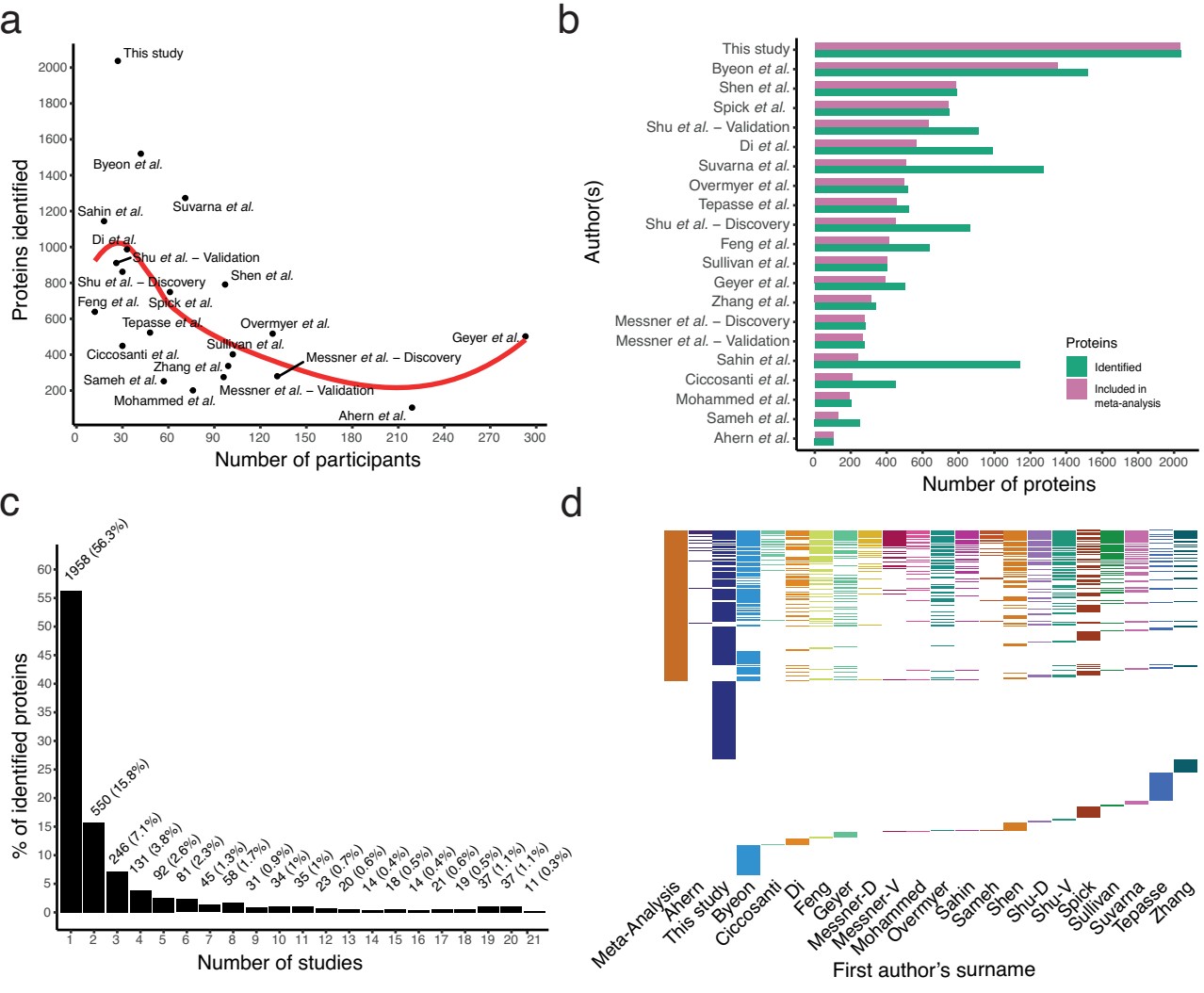

**Fig. 6 | Protein identification in studies included in the meta-analysis. a** Number of participants in relation to protein identification—as reported in the publication. The line was fitted with local polynomial regression; **b** Proteins identified compared to proteins included in the meta-analysis, i.e., proteins identified in at least three COVID-19 patients and at least three SARS-CoV-2 PCR-negative controls;

**c** Proportion of proteins identified in the studies. Only identifications that had a unique match to a protein with a gene name were included—those that had two gene names per protein were excluded; **d** Map depicting the overlap of proteins identified in a given study.

proteins. The SMD penalises the difference in levels for the variance in each group, and hence is a better estimate of change when combining studies with different methodological protocols. We applied both fixed-effects and random-effects models[51]. To infer how dissimilar the cohorts and quantifications are, we estimated the heterogeneity in the studies with the coefficients $Q$, $I^2$, and $\tau$. Because the heterogeneity was high, as estimated with $I^2$, we based our interpretation on the random-effects model.

Of the 1517 soluble blood proteins, a large proportion had a statistically significant alteration ($n = 338$, 22.28%, Fig. 7a, b, Supplementary dataset 13a). Comparing DAPs in serum identified by HiRIEF LC-MS/MS (two-sided $t$ test, $p < 0.05$, 5% FDR) to DAPs significant in the meta-analysis (random effects model, $p < 0.05$) showed a very high agreement in estimating the direction of the change, affirming our results (Fig. 7c). Stratifying the meta-analysis based on sample type showed no major differences between the findings based on studies performed on serum compared to those performed on plasma samples, when comparing the findings to HiRIEF LC-MS/MS (Figure S18, Supplementary Datasets 13b, c). Examples of SMD forest plots that are available for download from the CoViMAPP shiny app (https://doi.org/10.17044/scilifelab.22293148) are shown in Figure S19. We provided some level of modularity for the users of CoViMAPP, to stratify the per-protein meta-analyses based on sample type, acquisition type, and remove studies deemed as outliers.

The meta-analysis clarified the direction of protein alterations in instances of conflicting results, such as in the case of LBP, B2M, the antibody heavy chain variable domain IGHV3-23, and the proteasomal proteins PSMB5 and PSMB8 (Figure S19). E.g., we and Filbin et al.[16] found elevated blood levels of LBP in COVID-19, whereas SOMAscan analyses by Sullivan et al.[9] showed no change in serum levels. Including all cohorts in the meta-analysis of LBP showed increased LBP blood levels in COVID-19. Another example is B2M, a component of MHC class I molecules, which had variable estimates reported in the different MS studies. Pooling these estimates into one showed that B2M is elevated in COVID-19. An even more explicit example of the power of meta-analysis to detect alterations is that of the immunoglobulin heavy chain variable IGHV3-23, where all but two studies found no alterations in the protein levels, but the summary estimate showed increased levels in the blood. The meta-analytical estimates further reaffirmed our findings of elevated proteasomal proteins in the blood of COVID-19 patients, even though the few other studies that quantified the proteins found no change. E.g., although PSMB5 has been identified in three other studies, only this study identified it as elevated in the blood, which was further confirmed in the meta-analysis. Apart from these, the meta-analysis further clarified the direction of the alteration of other proteins with potential biological relevance in COVID-19, such as the decrease in the soluble blood levels of the receptors for the colony-stimulating factor 1 (CSF1R) and fibroblast growth factor (FGFR1), likely due to their internalisation after binding their ligands.

These are just few of the many examples that demonstrate the power of meta-analysis to capture changes that would not be picked up in a single or multiple studies. These cases exemplify the value of both the meta-analysis as an analytical method and CoViMAPP's availability as a resource to be used by the research community.

### SROC curves of soluble blood proteins

Finally, to estimate the potential of each soluble blood protein identified in at least three studies to differentiate COVID-19 status, we performed a summary receiver operating characteristics (SROC) meta-analysis with Reitsma et al.'s bivariate random-effects model[52,53]. The bivariate model provides summary estimates of sensitivity, specificity, and area under the curve (AUC), and is superior to univariate analyses of diagnostic odds ratios[54]. A total of 179 out of 971 proteins identified in at least three studies were useful in differentiating COVID-19 from SARS-CoV-2 PCR-negative controls (Supplementary Dataset 14a), some

of which showed high AUC (Fig. 7d, e). Most of the proteins that had a larger absolute value of SMD were also identified as the best discriminators between COVID-19 cases and SARS-CoV-2 PCR-negative controls. Some of these proteins had a similar or potentially better AUC performance than CRP, which is widely considered as one of the most sensitive and specific biomarkers of inflammation (Figure S20). To test whether the ROC curves of the underlying studies preferred sensitivity or specificity and how this affected the meta-analysis estimates of sensitivity and specificity, we analysed the ROC curve preference with the $\alpha$ parameter, following the approach by Doebler and Holling (2015)[55]. Although there was a positive association between higher values of $\alpha$, corresponding to study-specific ROC curves' preference for sensitivity, and higher values of sensitivity over specificity in the meta-analysis estimates, the effect was rather minor (Fig. 7f, Supplementary Dataset 14b). Only 2% of the variance in the log ratio of sensitivity over specificity was explained by the variance in the study-specific values of $\alpha$ in the underlying studies. As expected, there was a trade-off between sensitivity and specificity of the proteins in differentiating COVID-19 (Fig. 7g). The number of included cohorts had a minor effect on decreasing the absolute difference between sensitivity and specificity and no effect on the AUC estimates (Fig. 7h). This suggests that including more cohorts in the meta-analysis leads to a minor tendency for the sensitivity and specificity estimates to converge.

In summary, we performed a meta-analysis on COVID-19 soluble blood proteome alterations, estimating both the SMD and SROC curves per protein. By summarising our estimates with estimates from 20 other cohorts, we assert that a large proportion of the soluble blood proteome is altered in COVID-19 and report proteins that can differentiate COVID-19 from PCR-negative controls with high accuracy. In addition, we show that HiRIEF LC-MS/MS had very high categorical agreement with the SMD estimates in the meta-analysis for overlapping proteins, emphasising the method's accuracy and precision, apart from the method's potential for in-depth analysis of the soluble blood proteome.

## Discussion

Mass vaccinations against COVID-19 have substantially changed the course of the pandemic. However, COVID-19 is likely to remain a severe disease that can lead to long-term health effects or death in some individuals, making it a relevant disease to understand in a systematic manner. Partial understanding of the systemic host response in COVID-19, i.e., the cytokine storm, has already led to successful drug repurposing in COVID-19, such as corticosteroids and monoclonal antibodies against the IL6 receptor, both of which target immune-related proteins involved in inflammation and have a moderate improvement of survival[56–58]. We argue that discovery of biomarkers for diagnosis, prognosis, and treatment in COVID-19 requires an in-depth analysis of the soluble blood proteome. However, the soluble blood proteome has a high dynamic range of concentrations, where a small number of proteins account for most of the protein mass[59], making in-depth soluble blood proteomics by MS rather difficult. This is evident in our systematic review on COVID-19 soluble blood proteomics where the median number of proteins identified across the different MS proteomics platforms was 523, even though the soluble blood proteome is known to harbour at least ~4500 proteins[14], leaving less-abundant proteins undetected.

With this study we contribute to several advances in COVID-19 proteomics, pinpointing host responses to SARS-CoV-2 infection in human cells and blood serum. Our methodological and analytical approaches can be extended to profiling host responses in other infections. First, by quantifying ~2000 proteins, we provide the most in-depth soluble blood proteome coverage in COVID-19 by an MS method. Second, HiRIEF LC-MS/MS detected elevated levels of interferon- and proteasomal- proteins in serum of COVID-19

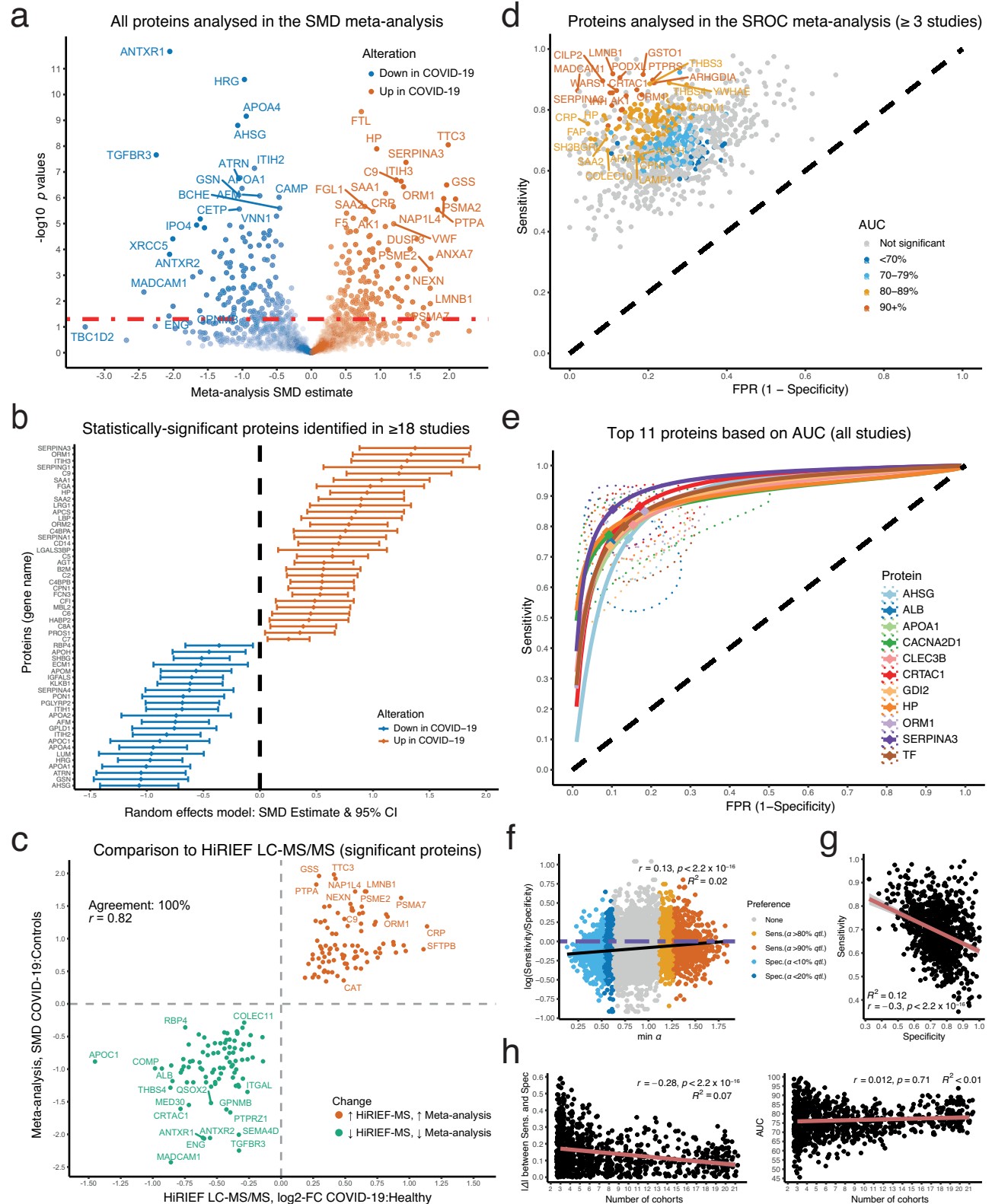

patients, which correlated to anti-SARS-CoV-2 immune response and markers of severity. We also report several tissue-enriched proteins altered in serum of COVID-19 patients, potentially deriving from the liver, lung, intestines, and brain, which could relate to multi-organ involvement. Apart from identifying previously undescribed alterations of soluble blood proteins, we provide further validation for dozens of soluble blood proteome alterations

identified with AB methods by detection of the protein's amino acid sequence. The categorical agreement on the direction of alteration between the overlapping statistically significant proteins identified by HiRIEF LC-MS/MS and the AB-methods was very high. However, there were still many proteins that were identified by one of the three methods, but not by the remaining two. This makes different proteomic platforms complementary and very relevant in co-

**Fig. 7 | Meta-analysis summary and selected proteins. a** Volcano plot showing the SMD on the x axis and -log10 *p* values on the y axis, random-effects model. Proteins above the dashed line (*p* < 0.05) were statistically significant. The shade of the points represents SMD multiplied by -log10 (*p* value); **b** Proteins identified in ≥ 18 cohorts that had a statistically significant SMD. The error bars represent 95% CI of the SMD estimates; **c** Agreement in percentages and Spearman's correlation coefficient (*r*) between log2-FC by HiRIEF LC-MS/MS (*p* < 0.05, 5% FDR) and SMD estimates in the meta-analysis (*p* < 0.05); **d** SROC summary estimates of sensitivity and specificity of all the proteins identified in ≥ 3 studies. Proteins with 95% CI of either sensitivity or specificity including 0.5 (the chance dashed line) were statistically non-significant. FPR false positive rate; **e** Top 11 proteins based on AUC that were identified in all cohorts. The SROC curves are based on the bivariate model, along with a 95% CI tolerance ellipsoid per protein; **f** Heterogeneity of the underlying ROC

curves per protein per study (*n* = 7606) used in estimating the SROC curves (*n* = 971). The estimated *α* shape parameter for lowest heterogeneity (*Q*) is plotted on the x axis; *α* of 1 = no preference, *α* > 1 indicates ROC preference for sensitivity and *α* < 1 indicates ROC preference for specificity. The log ratio of mean sensitivity and mean specificity is presented on the y axis; 0 = no preference (horizontal double-dashed line), values > 0 = model prefers sensitivity, and values < 0 = model prefers specificity. The ROC curves for a protein in a study were labelled as having a preference of sensitivity and specificity if they had values above the 80th quantile (*qtl*) and below the 20th quantile of *α*, respectively. **g** Inverse relationship between mean specificity (x axis) and mean sensitivity (y axis). **h** Relationship between the number of cohorts included in the per-protein estimates and absolute difference between specificity and sensitivity (left) and AUC (right).

validating findings, pinpointing differences that remain consistent across methods and across cohorts.

Third, following proteome alterations in vitro from an infection experiment at later time points after infection (day 3 and day 7), we find SARS-CoV-2 induced proteome changes at cellular level that have not been described in other studies, possibly due to short follow-up after infection[60]. We observed the highest levels of viral protein sequences at day 3, which could explain why there were no changes in the proteome at the earlier time points and why the best agreement with the protein alterations in serum levels in COVID-19 patients was to those occuring 3 days after infection. By integrating the cellular and serum proteomics datasets, we show that alterations in serum levels of proteins involved in interferon signalling, proteasome, and ubiquitination can relate to similar alterations after a SARS-CoV-2 infection in vitro. The serum proteome in severe COVID-19 showed the strongest increase in innate immune response, specifically in the NF-kB, IFN-α and IFN-γ pathways, and proteasomal proteins. Heptamers of proteasomal proteins build the α and β rings of the proteasome, a crucial organelle that maintains proteostasis by degrading ubiquitinated proteins, cleaves viral proteins into peptides that are to be presented as antigens through MHC class I molecules, and cleaves precursor proteins that activate the NF-kB pathway. The constitutional cellular proteasome consists of three catalytically active subunits—β1 (gene name: PSMB6), β2 (PSMB7), and β5 (PSMB5)—that regulate the proteostasis in healthy human cells[61]. However, alternate subunits of the same proteins—referred to as β1i (PSMB9), β2i (PSMB10), and β5i (PSMB8)—can be induced during immune response, particularly due to IFN-γ stimulation. These subunits assemble quickly during immune response into a specialised organelle—the immunoproteasome—and provide faster degradation of proteins into peptides, required for antigen presentation[62]. Apart from showing that different proteasomal proteins are elevated in the blood of COVID-19 patients, we could further differentiate that both the constitutional- and the immunoproteasome β subunits were elevated; specifically, the constitutional β1 (PSMB6) and β5 (PSMB5) subunits, and the immunoproteasome subunits β2i (PSMB10) and β5i (PSMB8). We observed a large increase in both immunoproteasomal subunits PSMB8 and PSMB10, and the IFN-activated proteins ISG15 and MX1 in serum and in infected cells 3 and 7 days after infection, thus connecting these changes in blood proteins to those after SARS-CoV-2 infection. Still, the Calu-3 cell line is an epithelial cancer cell line, which might limit the number of alternating proteins that can be traced in the blood. However, perhaps as challenging as claiming tissue specificity of the circulating proteins is delineating proteome alterations in infected organs from those induced by systemic processes such as inflammation. This posed as a challenge in previous proteome analyses of organs sampled from deceased COVID-19 patients in comparison to those of control donors by Schweizer et al.[30]. The authors have addressed this by annotating well-known plasma and immunoglobulin proteins, to differentiate them from organ-localised proteome alterations. Still, the question of how much the tissue proteomes were contaminated with the soluble

blood proteome remains open. In our study, we investigated proteome alterations occurring in an in vitro system of infection that is isolated from the systemic effects, providing insights that many of the organ-localised proteome alterations have been likely induced by SARS-CoV-2 infection. Furthermore, we show that a core set of 11 proteins seems consistently upregulated during SARS-CoV-2 infection in vitro, in COVID-19 patient serum and tissue samples, and that detected alterations in organ-associated protein sets can be traced in the blood. Our results pinpoint that the proteins decreasing during COVID-19 in the white pulp of the spleen[31], one of the largest immune organs in the body, might be due to their shedding in the bloodstream. It is worth reminding the reader that although we find similar protein alterations that occur in the serum of COVID-19 patients, after infection in vitro, and in different tissues obtained from deceased COVID-19 patients, these do not necessarily prove with certainty that the alterations are specifically and directly deriving from SARS-CoV-2 infection or a specific organ. However, the evidence supports the hypothesis that some of the observed proteome alterations in the blood of COVID-19 patients are likely derived from proteome alterations in cells and organs during SARS-CoV-2 infection, whereas some appear to be shared across different organs as a systemic host response to infection.

Furthermore, we identified phosphorylated proteins in the serum that had exact matching to peptides in Calu-3 cells infected with SARS-CoV-2 in vitro. Among them, we discovered that levels of phospho-calnexin (CANX S564-p) had decreased serum levels in COVID-19 patients, unlike the canonical non-phosphorylated protein. Calnexin is a molecular chaperone that has been implicated in regulating protein folding, in keeping misfolded glycoproteins in the endoplasmic reticulum (ER)[63]. Furthermore, it is reported to interact with SARS-CoV-2[64] and plays a role in stabilising the α chains of MHC class I molecules prior to the binding of β-2 microglobulin, thus being involved in regulating antigen presentation[65]. Although phosphorylations of the protein have been described before[66], not much is known about their functionality. Phosphorylation of calnexin through the MEK-ERK1 pathway has been associated with attenuated release of misfolded proteins, such as α-1-antitrypsin (SERPINA1), where inhibiting the phosphorylation has led to an increase in secretion of misfolded proteins[67]. Considered that non-phosphorylated levels of calnexin have shown an increase during SARS-CoV-2 infection in vitro in this study and in vivo in previous studies[30,31], it is enticing to speculate that decreased phosphorylation of calnexin identified in the blood in our study and by others in lung tissue[30] leads to impaired quality control of protein folding in the ER, permitting secretion of misfolded proteins that avoid degradation by the proteasome complex.

Lastly, the in-depth coverage by HiRIEF LC-MS/MS and the availability of 20 additional MS datasets allowed us to perform a meta-analysis on COVID-19 global soluble blood proteomic datasets, which provides the highest level of evidence on blood levels' alterations of ~1500 proteins and the diagnostic potential of almost a thousand proteins in ~1700 individuals. Only two studies[49] included a validation global proteomics cohort comparing COVID-19 to SARS-CoV-2

negative controls, making this meta-analysis an important independent validation resource for findings reported in individual discovery studies. However, the value of the meta-analyses is not only in validating discovered findings, but in further detecting alterations that are not observed in single studies and deriving conclusions in instances where the results are conflicting, as exemplified with LBP, B2M, IGHV3-23, PSMB5, CSF1R, etc. There is an abundance of conclusions that can be inferred from the meta-analysis that are beyond the scope of this manuscript's discussion. Therefore, we further provide these results on soluble blood proteome alterations in COVID-19 as CoViMAPP (https://doi.org/10.17044/scilifelab.22293148), a publicly available resource for the research community. Although we did not remove outliers when interpreting the results of the meta-analysis, the CoViMAPP users might wish to do so, to provide a better estimate for the protein of interest. Traditionally, CRP has been used as a systemic clinical marker of inflammation in COVID-19 and other diseases, although the SROC curves suggest that other acute phase proteins, such as ORM1, CRTAC1, SERPINA3, TF, and AHSG might perform better as biomarkers of inflammation. This indicates that maybe some acute phase proteins that are repeatedly identified with global MS proteomics in the soluble blood proteome should be reconsidered for clinical use.

As expected, there was a large heterogeneity between the studies, which can be driven by many factors: accuracy and precision of methods, gene/protein mapping, selection bias, adjustment for different clinical factors, data normalisation, genetic and ethnic differences, different SARS-CoV-2 variants, COVID-19 severity, choice of controls, etc. All of these should be considered when interpreting the results. Whereas this study provides the current MS-based state-of-the-art concluding estimates for soluble blood proteome alterations in COVID-19, we envision CoViMAPP as a dynamic meta-analysis resource, which will analytically curate future studies and include them in the summary estimates.

In summary, we report a comprehensive analysis of the soluble blood proteome alterations in COVID-19 by MS proteomics, where we find alterations of soluble blood proteins that add to our understanding of COVID-19 pathogenesis, and validate several previous findings reported by AB methods. We demonstrate that soluble blood proteome alterations can be traced to SARS-CoV-2-infected cells. Finally, by performing a comprehensive meta-analysis of soluble blood proteome alterations in COVID-19 quantified by MS methods, we developed CoViMAPP, an open-access resource that is available for the scientific community to aid future research on soluble blood proteomics and circulating biomarkers in COVID-19.

## Methods

### Patients

Twenty hospitalised SARS-CoV-2 PCR-confirmed patients from Karolinska University Hospital in Stockholm, Sweden, were included in the study in April 2020. Data from the cohort and methodology of clinical and immunological assays has been previously described in detail elsewhere[24]. Inclusion criteria were ongoing acute COVID-19 disease and self-declared healthy individuals, for cases and controls, respectively. Exclusion criteria for both groups were known immunosuppression or immunosuppressive disease. Serum was collected from COVID-19 patients and healthy controls in BD Vacutainer serum tubes with spray-coated silica (BD Biosciences). After coagulation for up to 2 hours at room temperature, serum was isolated by centrifugation at $2000 \times g$ for 10 min and immediately stored at $-80\,°C$ for later analysis.

The study was approved by the Regional Ethical Review Board in Stockholm, Sweden and by the Swedish Ethical Review Authority, and is in accordance with the Declaration of Helsinki. All COVID-19 patients and healthy controls included in this study provided written informed consent to participate in the study.

### In vitro infection experiment

Human lung adenocarcinoma Calu-3 cells (ATCC, HTB-55) were grown in minimum essential medium (MEM) supplemented with 20% FBS, HEPES, L-glutamin, 100 U/ml penicillin, and 100 mg/ml streptomycin at $37\,°C$ and 5% $CO_2$. The SARS-CoV-2 ancestral variant (isolate SARS-CoV2/human/SWE/01/2020; Genbank accession: MT093571) was propagated on Vero E6 cells and titrated via end-point dilution assay. The AV was chosen as this was the SARS-CoV-2 strain that the COVID-19 patients had been infected with. For UV-inactivation, the virus stock was incubated under UV-light for $3 \times 1.5$ min. Complete inactivation was confirmed via infection attempt and no live virus could be detected after the treatment.

Cells were infected with active or UV-inactivated SARS-CoV-2 in complete MEM at a multiplicity of infection (MOI) of 1. After two hours of incubation the virus solution was removed, the cells were washed, and fresh growth medium was added. Before the infection, the active and inactivated virus were treated with Trypsin at 1:100 dilution for one hour at $37\,°C$. Before sample collection at the indicated time points, the cells were washed thoroughly.

### Mass-spectrometry serum proteomics

**High abundant protein depletion.** Depletions were performed using the High Select™ Top14 Abundant Protein mini columns (Thermofisher), according to manufacturer's recommendations. Briefly, 10 μL of serum were applied to each Mini column and incubated at room temperature with gentle end-over-end mixing, for 20 min. Post depleted samples were then heated at $56\,°C$ for 30 min for viral inactivation. Depleted flowthroughs were recovered by centrifugation. The depleted serum flow-through was concentrated on 5 kDa molecular weight cut-off filter followed by buffer exchange to 50 mM HEPES pH 7.6, as previously described[68].

**MS Sample preparation.** Depleted serum was denatured at $60\,°C$ for 1 hour followed by reduction with DTT at $95\,°C$ for 30 min and alkylation with chloroacetamide at room temperature for 20 min at end concentrations of 4 mM. Trypsin was added at a 1:50 (w/w) ratio and digestion was performed at $37\,°C$ overnight. TMT-16 labelling was performed according to manufacturer's instructions and labelling efficiency was evaluated by LC-MS/MS on pooled samples using 30 min gradients to ensure >95% labelling of peptides before pooling. After pooling the samples, 1 mL Strata X-C 33 μm columns (Phenomenex) were used for sample clean-up. The peptides were subsequently dried in a SpeedVac. HiRIEF separation was performed, as previously described[69,70]. Briefly, the samples were rehydrated in 8 M urea with bromophenol blue and 1% IPG buffer, and subsequently loaded to the immobilized 3–10 pH gradient (IPG) strip and run according to previously published isoelectric focusing (IEF) protocols[69]. After IEF, the IPG strip was eluted into 72 fractions using an in-house robot. The obtained fractions were dried using SpeedVac and frozen at $-20\,°C$ until MS analysis.

**LC-ESI-MS/MS Q-Exactive HF.** Q-Exactive Online LC-MS was performed using a Dionex UltiMate 3000 RSLCnano System coupled to a Q-Exactive-HF Hybrid Quadrupole-Orbitrap mass spectrometer (Thermo Scientific). Each of the 72 plate wells was dissolved in 20 μL solvent A and 10 μL were injected. Samples were trapped on a C18 guard-desalting column (Acclaim PepMap 100, 75 μm × 2 cm, nanoViper, C18, 5 μm, 100 Å), and separated on a 50 cm long C18 column (Easy spray PepMap RSLC, C18, 2 μm, 100 Å, 75 μm × 50 cm). The nano capillary solvent A was 94.9% water, 5% DMSO, 0.1% formic acid; and solvent B was 4.9% water, 5% DMSO, 90% acetonitrile, and 0.1% formic acid. At a constant flow of 0.25 μL min$^{-1}$, the curved gradient went from 6 to 10% B up to 40% B in each fraction in a dynamic range of gradient length, followed by a steep increase to 100% B in 5 min. Information on gradient length is provided in Supplementary dataset 15.

Spray voltage was set to 1.9 kV, S-lens RF level at 60, and heated capillary at 275 °C. Full scan target was $3 \times 10^6$ with a maximum fill time of 15 ms. FTMS master scans with 60,000 resolution (and mass range 300–1500 m/z) were followed by data-dependent MS/MS (30,000 resolution) on the top 5 ions using higher energy collision dissociation (HCD) at 30% normalized collision energy. Precursors were isolated with a 2 m/z window. Automatic gain control (AGC) targets were $1 \times 10^6$ for MS1 and $1 \times 10^5$ for MS2. Maximum injection times were 100 ms for MS1 and 400 ms for MS2. The entire duty cycle lasted ~2.5 s. Dynamic exclusion was used with 30 s duration. Precursors with unassigned charge state or charge state 1 were excluded. An underfill ratio of 1% was used. All data were acquired in positive polarity mode.

## Mass-spectrometry cell proteomic analysis of cells

The cell pellets were dissolved in 300 µl of lysis buffer (4% SDS, 50 mM HEPES pH 7.6, 1 mM DTT), heated to 95 °C, and sonicated. The total protein amount was estimated with the Bio-Rad DC assay. Samples were then prepared for MS analysis using a modified version of the SP3 protein clean-up and a digestion protocol[71,72], where proteins were digested by Lyc-C and trypsin (sequencing grade modified, Pierce). In brief, 200 µg protein from each sample was alkylated with 4 mM chloroacetamide. Sera-Mag SP3 bead mix (20 µl) was transferred into the protein sample together with 100% acetonitrile to a final concentration of 70%. The mix was incubated under rotation at room temperature for 30 min. The mix was placed on the magnetic rack and the supernatant was discarded, followed by two washes with 70% ethanol and one with 100% acetonitrile. The beads-protein mixture was reconstituted in 100 µl Lys-C buffer (1 M Urea, 50 mM HEPES pH: 7.6 and 1:50 enzyme (Lys-C) to protein ratio) and incubated overnight. Finally, trypsin was added in 1:50 enzyme to protein ratio in 100 µl 50 mM HEPES pH 7.6 and incubated overnight. The peptides were eluted from the mixture after placing the mixture on a magnetic rack, followed by peptide concentration measurement (Bio-Rad DC Assay). The samples were then pH adjusted using TEAB pH 8.5 (100 mM final concentration), 100 µg of peptides from each sample were labelled with isobaric TMT tags (TMT-16 reagent) according to the manufacturer's protocol (Thermo Scientific), and the pooled peptide mix was further fractionated by basic reverse phase chromatography. In particular, the separation was performed using a 25 cm column (Waters corporation, XBRIDGE, Peptide BEH C18 column, 300 Å, 3.5 µm, 2.1 mm × 250 mm) in a 90 min gradient from 3% solvent A (20 mM NH₃) to 50% solvent B (80% Acetonitrile, 20 mM NH₃) at a constant flow of 200 µl/min. Finally, 96 fractions were collected, concatenated in 24 as previously described[73], and analyzed by LC-MS/MS. The LC-MS/MS analysis was performed using a Dionex UltiMate™ 3000 RSLCnano System coupled to a Q-Exactive-HF mass spectrometer (Thermo Scientific). Each of the 24 fractions was dissolved in 20 µl solvent A and 10 µl were injected. Samples were trapped on a C18 guard-desalting column (Acclaim PepMap 100, 75 µm × 2 cm, nanoViper, C18, 5 µm, 100 Å), and separated on a 50 cm long C18 column (Easy spray PepMap RSLC, C18, 2 µm, 100 Å, 75 µm × 50 cm). The nano capillary solvent A was 94.9% water, 5% DMSO, and 0.1% formic acid; and solvent B was 4.9% water, 5% DMSO, 90% acetonitrile, and 0.1% formic acid. At a constant flow of 0.25 µl min⁻¹, the curved gradient went from 6–8% B up to 40% B in each fraction in a dynamic range of gradient length, followed by a steep increase to 100% B in 5 min.

FTMS master scans with 60,000 resolution (and mass range 300–1500 m/z) were followed by data-dependent MS/MS (30,000 resolution) on the top 5 ions using higher energy collision dissociation (HCD) at 30% normalized collision energy. Precursors were isolated with a 2 m/z window. AGC targets were $1^6$ for MS1 and $1^5$ for MS2. Maximum injection times were 100 ms for MS1 and 400 ms for MS2. The entire duty cycle lasted ~2.5 s. Dynamic exclusion was used with 30 s duration. Precursors with unassigned charge state or charge state 1 were excluded. An underfill ratio of 1% was used.

Of note, the labelling efficiency was determined by LC-MS/MS before pooling of the samples. For the sample clean-up step, a solid phase extraction (SPE strata-X-C, Phenomenex) was performed, and purified samples were dried in a SpeedVac. An aliquot of approximately 10 µg was suspended in LC mobile phase A and 1 µg was injected on the LC-MS/MS system.

## Peptide and protein identification and quantification

Orbitrap raw MS/MS files were converted to mzML format using msConvert from the ProteoWizard tool suite (v.3.0.20066)[74]. Spectra were searched using the MSGF+ search engine (v2020.03.14)[75] and Percolator (v3.04.0)[76] for target-decoy scoring.

All searches were done against the Human protein coding subset of Ensembl (v.105) using our proteomics workflow (https://github.com/lehtiolab/ddamsproteomics, v.2.7), which was run with Nextflow (v.20.01.0). MSGF+ settings included precursor mass tolerance of 10 ppm, fully tryptic peptides, maximum peptide length of 50 amino acids and a maximum charge of 6. Fixed modifications were carbamidomethylation on cysteine residues and TMT-16 on lysine and peptide N-termini. A variable modification was oxidation on methionine residues.

Quantification of TMT-16 reporter ions was done using the OpenMS project's IsobaricAnalyzer (v.2.5)[77]. MS1 feature detection and quantification was performed using Dinosaur (https://github.com/fickludd/dinosaur)[78]. PSMs found at 1% FDR were used to infer gene identities. Protein quantification by TMT-16 reporter ions was calculated using medians of log2-transformed PSM channel intensities from which the values of the internal standard(s) were subtracted. Protein and gene quantification values were then normalized by subtracting their channel medians. Protein false discovery rates were calculated using the picked-FDR method using gene symbols as protein groups and limited to 1% FDR[79].

## Phosphoproteomics

The raw MS files from the serum analysis were researched again, matching MS spectra to the human Ensembl (v.105) protein database, with the same search parameters as specified above, additionally specifying phosphorylation of serine, threonine, or tyrosine as a variable modification.

The samples from the SARS-CoV-2 infection experiment were enriched for a phosphoproteomics analysis. Briefly, cell pellets were lysed by SDS lysis buffer (25 mM HEPES, pH: 7,6, 4% SDS, 1 mM DTT, 1% PMSF) and prepared for mass spectrometry analysis using a modified version of the SP3 protein clean up and digestion protocol[71] and a modified protocol for phosphoproteomics[73]. Peptides were labelled with TMT-16 reagent according to the manufacturer's protocol and separated by basic pH Reverse Phase Chromatography. The concatenated fractions from the prefractionation were subjected to phosphopeptide enrichment using Ti-IMAC chromatography. The phospho-enriched samples as well as non-enriched (for total proteome characterisation) samples were separated in a 120-min gradient using an online 3000 RSLCnano system coupled to a Thermo Scientific Q Exactive-HF. MSGF+ and Percolator in the Galaxy platform was used to match MS spectra to the human Ensembl (v.105) protein database, using phosphorylation of serine, threonine, or tyrosine as a variable modification.

## Data analysis

**Statistics and reproducibility.** No statistical method was used to predetermine sample size. No data were excluded from the analyses. The experiments were not randomized. The investigators were not blinded to allocation during experiments and outcome assessment.

**Descriptive statistics and differential analysis.** Descriptive statistics was reported as mean, median, and standard deviation for numeric variables, and percentages for categorical variables.

The serum proteomics data have been previously normalised to the mean of two and three internal standards in set 1 and 2, respectively. The data were further median centred and normalised with a log2-transformation. A univariate differential analysis of serum protein levels was performed with a two-sided $t$ test at a significance level of $\alpha = 0.05$ and corrected for multiple testing with the FDR. The comparison was further adjusted in a multivariate limma model and two-sided modified $t$ statistic, adjusting for age, sex, hypertension, and diabetes[25], and corrected for multiple testing with the FDR. The log2-FC refers to the log2 mean difference between serum protein levels in COVID-19 and healthy controls. The agreement between HiRIEF LC-MS/MS estimates of the log2-FC and log2-FC estimated by PEA and SOMAscan was calculated categorically, as percentage of proteins with a change in the same direction and a Spearman correlation coefficient between the log2-FC estimates of the two studies. In the main analysis, we compared only the proteins that were statistically significant in both compared methods at p < 0.05 and 5% FDR. In two separate sensitivity analysis comparison to PEA and SOMAscan data, we also compared the agreement between log2-FC of proteins that had a statistically significant alteration in HiRIEF LC-MS/MS at $p < 0.05$ and 5% FDR to the log2-FC in the other method that was significant at $p < 0.05$ or non-significant.

**Serum protein clustering.** We clustered the patients based on normalised protein relative expression values of proteins without missing values with the principal component analysis (PCA) and hierarchical clustering. For hierarchical clustering, we used Spearman's correlation coefficient as a distance metric $(1 - r)$ and presented the protein expression values in a heatmap. The enrichment terms presented next to the heatmap were derived from a two-sided overrepresentation test on the corresponding protein clusters obtained through hierarchical clustering.

**Tracing serum proteins to SARS-CoV-2-infected organs.** Data on DAPs in SARS-CoV-2-infetected organs were obtained from proteomics datasets published by Schweizer Lisa et al.[30] and Xie Nie et al.[31] Both datasets contained lists of DAPs per SARS-CoV-2-infected tissue or organ as compared to matching non-SARS-CoV-2-infected tissue or organ controls. Schweizer et al. analysed ten different organs, including 11 tissue types: lungs, lymph nodes, liver, adrenal gland, kidney, *medulla oblongata* (brain), basal ganglia (brain), blood vessels, walls of blood vessels, heart, and spleen. Nie et al. analysed seven different organs, including 9 tissue types: lungs, red pulp of the spleen, white pulp of the spleen, liver, heart, renal cortex, renal medulla, testis, and thyroid. Each study-specific per-organ protein list of DAPs was divided into two protein sets, one consisted of upregulated DAPs, and another consisted of downregulated DAPs, totalling 40 protein sets. The protein sets were annotated with the first author's initials, the organ type (and tissue, where applicable), and the direction of the change (-UP/-DOWN), e.g., SL Lungs UP, XN Liver DOWN, etc. These lists were then intersected with DAPs in the serum or cell lines identified in this study and annotated in protein maps, as described in supplementary figures.

To detect enrichment in the serum of COVID-19 patients of proteins detected in SARS-CoV-2-infected organs we performed GSEA for each protein set that had at least 10 proteins in the set ($n_{sets} = 33$), after filtering out any protein classified as a plasma protein or immunoglobulin[30], and any protein with log2-FC < 0.5. The statistical significance in the GSEA was determined with the permutation test and the $p$ values corrected for multiple testing with the FDR. The GSEA was performed at 5% FDR and Entrez IDs for the protein sets were extracted with the bitr function in clusterProfiler[80]. To increase the tissue specificity, in an additional analysis we have filtered out any protein in an

organ-associated set that has been a DAP in any other organ-associated set in the same study. We then performed another GSEA analysis on the filtered protein sets, if they included at least 10 proteins ($n_{sets} = 33$).

**Enrichment analyses.** All enrichment analyses were performed with the package clusterProfiler[80]. The gene sets were fetched from the MSigDb database. The overrepresentation test in the heatmap was performed with a two-sided Fisher's exact test at $\alpha = 0.05$ and 5% FDR, using all identified proteins as background. GSEA[32] was performed as previously described, per groups of gene sets as classified in the MSigDb[33] at 5% FDR, using all the proteins quantified in at least 50% of the observations and converting gene name IDs to Entrez IDs. The ranking of the proteins was based on the mean difference between serum levels in COVID-19 patients and serum levels in healthy individuals.

**Cellular proteomics differential analysis.** The cellular proteome changes attributable to infection were derived from a comparison between the infected Calu-3 cells with the SARS-CoV-2 AV at days 3 and 7 to the non-infected controls at days 3 and 7, respectively. The statistical comparison between the triplicates of infected cells and non-infected controls was performed with an unpaired two-sided $t$ test at a significance level of $\alpha = 0.05$. The agreement between the cellular protein log2-FC and serum log2-FC was calculated categorically, as percentage of proteins with a change in the same direction and a Spearman correlation coefficient.

**Differential abundance analysis of phosphorylated peptides.** Differential abundance of phosphorylated peptides between groups were tested with an unpaired two-sided $t$ test, at a significance level of $\alpha = 0.05$, only on phosphorylated peptides that were identified at 1% FDR and 5% FLR, observed in >50% and 100% of the serum and cell line samples, respectively. The phosphosite locations were mapped based on Ensembl canonical proteins for all peptides. The phosphosite locations were additionally mapped based on the Uniprot canonical proteins, if identified in the PhosphoSitePlus database (https://www.phosphosite.org/); if available, these IDs were used in the plots and in the text, otherwise the Ensembl phosphosite locations were annotated.

## Systematic review

On 15-02-2023, we searched for studies of interest in two databases, i.e., PUBMED and EMBASE, for the keywords: "(covid-19 OR sars-cov-2) AND (plasma OR serum) AND mass spectrometry AND (proteomics OR protein*)", without restriction. The references were handled in Mendeley reference manager. After excluding the duplicates, three authors (HB, NS, and JK) screened the articles manually based on inclusion and exclusion criteria. The inclusion criteria were studies with a global MS method on plasma or serum proteomes in human COVID-19 patients and PCR-negative human controls. We excluded reviews, conference articles, targeted proteomics studies, and studies performed on pregnant women or children. Studies selected in the screening step were then further evaluated for inclusion in the review and meta-analysis based on reading the full papers and accessing the publicly available processed normalised data from proteomic searches. Studies without available processed data were not included. Studies that had pooled samples before the MS analysis were excluded from the meta-analysis because the calculation of standard deviation would not have been valid. Details on how the received datasets were processed are available on github.

## Meta-analysis

After selecting the studies to include in the meta-analysis, for both the COVID-19 group and the SARS-CoV-2 PCR-negative control group we calculated the per-protein mean, standard deviation, and number of

persons in which the corresponding protein was identified. The quantifications for proteins with different Uniprot IDs that mapped to the same gene name were averaged and the quantification assigned to the gene name; if one of the proteins mapped to several Uniprot IDs, it was excluded prior to calculating the mean. Proteins that did not have a matching gene name ID were summarised based on their Uniprot ID. Details on the data processing are available on github. If a protein has been identified in at least three individuals in both groups, it was chosen for inclusion in the meta-analysis. Based on the mean, standard deviation, and number of participants, we estimated a SMD for each included protein in each study. We then pooled the SMD estimates in a fixed-effects and random-effects model and calculated a summary SMD estimate with 95% CI, a prediction interval, and measures of heterogeneity - $\tau$, $Q$ and $I^2$ coefficients[51]. The estimates were then plotted in a forest plot and the publication bias visualised with a funnel plot. The summary estimates with a 95% CI that did not overlap 0 were considered statistically significant.

We estimated the derived curves and summary sensitivity and specificity per protein with a modified approach of Reitsma et al.'s bivariate random-effects' model on contingency tables[52], available in the R package mada[53]. The approach uses a generalised linear mixed model for estimating sensitivity and specificity, utilising the glmer function from the lme4 package[81], as implemented by Sehovic et al.[82]. The contingency tables contained information on true positives, false positives, true negatives, and false negatives. The values were derived from selecting an optimum cut-off for a ROC curve with cutpointr, based on maximum Youden's index, for each protein, in each study. Then, an individual was categorised based on the measured protein levels as COVID-19 if their measured value was ≥ or ≤ the cut-off, when the protein was higher or lower in COVID-19, respectively. If the contingency tables had a zero in any of the cells, we added a correction coefficient of 0.5 to all cells. We calculated 95% CI for both sensitivity and specificity as well as coordinates for the 95% tolerance ellipsoid of the bivariate model. Only proteins that have been identified in at least three studies and at least three individuals in both groups were selected for inclusion in the meta-analysis. Following the approach by Doebbler & Holling (2015)[55], as implemented in[82], the heterogeneity of the underlying study-specific ROC curves was estimated with the $\alpha$ and C1 parameters, which indicate the ROC curves' preference for sensitivity/specificity and false positives/false negatives, respectively. Briefly, values of $\alpha = 1$ indicate no preference for sensitivity or specificity, whereas values of $\alpha > 1$ and $\alpha < 1$ indicate ROC curve's preference for sensitivity and specificity, respectively. Likewise, values of C1 > 1 and C1 < 1 indicate a preference for false positives and false negatives, respectively.

## Software
All analyses were performed in R (v.4.2.2), and the figures assembled in Adobe Illustrator 2023 (v.24.0.1). Figure 1 was created with Biorender (cloud-based software, last version accessed 15. July 2023). The list of used packages is provided in the code submitted to github.

## Reporting summary
Further information on research design is available in the Nature Portfolio Reporting Summary linked to this article.

## Data availability
The personal data are not publicly available due to them containing information that could compromise research participant privacy. All other data are provided in the article and its Supplementary files or from the corresponding authors upon request. Source data are provided with this paper. All results described in the manuscript are presented in main or supplementary figures or datasets. All results from the meta-analysis are available as a shiny app resource at https://doi.org/10.17044/scilifelab.22293148. Raw and processed MS data are

deposited via ProteomeXchange to the PRIDE database, with accession codes PXD037486, PXD037451, and PXD040982. Source data are provided with this paper.

## Code availability
The custom code for the Nextflow proteomics workflow is deposited at https://github.com/lehtiolab/ddamsproteomics. The R code used for the analyses is deposited at https://github.com/harbab/covid19proteomics.

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

## Acknowledgements

We acknowledge support of the Clinical Proteomics Facility and the Data Centre at Science for Life Laboratory. We express deep gratitude to Assistant Professor Rozbeh Jafari for providing us with server space and to Dr. Luay Aswad for helping with the infrastructure to transfer the CoViMAPP shiny app to a server. The project was funded by grants from the Knut and Alice Wallenberg foundation (KAW), within the SciLifeLab KAW COVID-19 national research program (ID: 2020.0182, to J.L., M.P., and J.K.), the Swedish Research Council (IDs: 2020-06249 and 2021-06602, to S.G.R.), Marianne and Marcus Wallenberg Foundation (SGR), Region Stockholm (clinical research appointment, to SGR), and by Centre for Innovative Medicine (S.G.R. and J.K.).

## Author contributions

Funding, conceptualisation, project administration, and supervision: J.L., J.K., and M.P. Clinical data and patient sample processing: R.V., M.G., H.A., H.G., and S.G.R. In vitro infection experiment: W.C. and J.T. Methodology: H.B., W.C., J.E.A., G.M., J.L., J.K., and M.P. Analysis: H.B., J.E.A., and G.M. Formal statistical analysis: H.B. Visualisation: H.B. Systematic review: H.B., N.S., and J.K. Meta-analysis and CoViMAPP design: H.B. Writing—original draft: H.B. Writing—methods: H.B., W.C., J.E.A., and G.M. Writing—final draft: all authors.

## Funding

## Competing interests

The authors declare no conflict of interests.
