## [Peer Review File · Nature Communications]

Reviewers' comments:

Reviewer #1 (Remarks to the Author):

This investigation by Babačić et al describes a mass-spectrometry based analysis of COVID-19 patient serum, in which they attempt to determine altered biomarkers associated with disease severity. These biomarkers were then compared to findings in an in vitro model of SARS-CoV-2 infection to confirm any alterations were infection based. They also describe a meta-analysis, in which the authors have compared 11 mass spectrometry-based databases in order to compare findings and identify commonly occurring biomarkers. This is a well written manuscript and describes an interesting methodology for maximizing the number of identifiable proteins detectable in serum.

Whilst, the manuscript describes a an in-depth approach to characterising the Covid-19 infected proteome the findings are not novel as many other studies that have done this and are quoted and referenced by the authors themselves. It would be better to see a further application of the methods that can give a unique biological insight about covid19 infection such as comparison with other infections, effect of prior vaccination or prediction of long term symptoms.

In addition, the experimental design is lacking of sample numbers for robust statistical analysis. It is appreciated the profiling method is long and intensive and the authors have tried to address this by validating using cell work and meta-analysis of other studies but these only confirm what is already described thereby lacking in novel findings. It would of been better to see any novel significant proteins/pathways identified of interest confirmed on much larger sample numbers using a targeted method such as ELISA or MRM-LC-MS/MS. This manuscript in its current form would be better revised to be a methodological study in another journal. For these reasons I do not recommend this manuscript suitable for publication in Nature communications.

Minor concerns

1. For the the meta-analysis comparing serum and plasma proteomic datasets – is this appropriate for two distinct biofluids? Sensitivity can be much greater in serum due to lower amounts of high abundant proteins in plasma, coupled with the Top 14 depletion that has been carried out in the serum analysis in this investigation. This should be clarified.
2. The authors seem to use serum and plasma proteome interchangeably. They have analysed serum proteome themselves but refer to changes in plasma proteome.
3. Methods, Solvent A detailed several times as 5% water, 95% acetonitrile, 0.1% formic acid, totalling 100.1%. Solvent B detailed several times as 5% water, 5% DMSO, 95% acetonitrile, 0.1% formic acid totalling 105.1%.
4. Mass spectrometer conditions need to be more detailed (source temp etc)
5. Intermittently use μ L instead of μ L throughout the methods section

Reviewer #2 (Remarks to the Author):

The manuscript analyzed the proteome of serum samples collected from 20 hospitalised patients with COVID-19 infected with the ancestral SARS-CoV-2 variant, and 7 healthy controls, which were PCR-negative and seronegative, using HiRIEF-based extensive fractionation and TMT-based quantitative proteomics. By performing a deep fractionated (72 fractions per batch, LC gradient unknown) TMT-based proteomic profiling, the author identified for more than 2000 proteins, which is deeper compared with most other LC-MS-based proteomics studies. Then they performed bioinformatics analysis of the proteomics data and tried to analyze the tissue origin of plasma proteins, but these bioinformatic analyses are descriptive and hypothetical. So does the comparison with PEA and SOMAscan data sets. Then the authors performed an in vitro cell line proteomic study with a deep proteomic coverage and claimed this could be used to gain insights into which proteins derived from SARS-CoV-2 infection site. This does not make much sense to the reviewer, because cellular proteome of cell line data can be hardly associated with circulating proteins in COVID-19 patients. With a FDR <5%, the authors performed bioinformatic analysis and discussed some of the regulated proteins when the cell line was perturbed by SARS-CoV-2. Finally, the authors claimed via meta-analysis with existing proteomic data sets, they explored the potential of selected proteins for COVID-19 diagnosis.

Major issues:

1, In general, this is a comprehensive proteomics study in terms of proteomic depth and bioinformatic analysis. However, the main contribution of this paper to the field is unclear. The patient sample size is relatively small, and the clinical value is minimal. The vague term “diagnosis” in the last section does not make much sense to the reviewer, because it is not a clinical need to diagnose COVID-19 by blood proteins. If the authors are to show that a protein-based model outperforms existing methods for diagnosing COVID-19 in the clinic, more experimental data are required. Differentiation between severe and non-severe COVID-19 cases is a clinical need, but multiple papers have been published in the field. This manuscript does not show its uniqueness in this regard neither.

2, The virus-infected cell line proteomic experiments should be applauded, but its association to the plasma proteome data set seems weak, if not far-fetched. The meta-analysis is useful but did not bring much new insights nor clinical value.

Minor issues:

1. More technical details for the procedures for tissue-enrichment analysis should be provided. Why only 1779 out of 2037 proteins were used for HPA enrichment analysis?

2, The statement of first meta-analysis of COVID-19 plasma proteome as well as its comprehensiveness might be over-claimed. Though described in the method section and Figure S10, it is not clear how the studies were collected and how the datasets were systematically normalized and curated. Please add more details so that this section could be potentially reproduced.

3, Have the authors provided the LC-gradient length for the LC-MS analysis? Not clear in the sections “LC-ESI-MS/MS Q-Exactive HF” and “Mass-spectrometry cell proteomics”.

4, Please provide more details of the criteria for inclusion and exclusion of a patient samples for this specific study, especially for healthy controls.

5, Please provide more details for the shiny app and provide a user manual.

6, "Automatic gain control (AGC) targets were 1×10^6 for MS1 and 1×10^5 for MS2." --- should be 10^{*6} and 10^{*5} (superscript).

7, Subtitle: "Mass-spectrometry cell proteomics": may be changed to Mass spectrometry-based proteomics analysis of cells

Reviewer #3 (Remarks to the Author):

Review of "Comprehensive proteomics and meta-analysis of COVID-19 host response detects elevated proteasomal proteins in blood traceable to SARS-CoV-2 infection"

I am not an expert on proteins (not even close). I have been asked to contribute a review of the statistical methodology and since I am not working in bioinformatics, the focus of the review will be on the meta-analytical methodology.

The manuscript studies proteins in persons with and without a CoViD infection using mass spectrometry. It has three main parts: (i) An original study with relatively small sample size but a large number of proteins, (ii) an in vitro experiments with infected cells, and (iii) an elaborate meta-analysis that adds data from other mass spectrometry studies. Given my own specialization, I have only two comments on (i) and (ii) but I can provide detailed feedback on (iii). Overall, the meta-analytical strategy of providing analyses for such a large number of biomarkers is laudable, but as implemented, it fails to leverage the complete potential of the data (see below).

Major Comments

=====

1) In your original study and elsewhere you use the customary t-tests and an FDR of 5% or 10% to screen for relevant proteins. The problem with the t-test, especially in the small sample size situation with 20 CoViD and 7 controls, is that we can never really check any distributional assumptions. Asymptotic normality is questionable, too. There might be proteins with small absolute mean differences but due to

underlying skewness or random chance, the sample variance is low in both groups. This in turn reduces the denominator in the t-statistic, inflating p-values. I recommend contrasting a more conservative procedure, e.g., the empirical Bayes approach in the LIMMA package (not sure if it works off-the-shelf though for MS data). I understand that this creates some work downstream, but since meta-analyses like these have the potential to point others in the wrong direction, a somewhat more conservative screening is advisable given the potential impact.

2) The diagnostic meta-analysis you perform takes the raw data, builds a fourfold table from that at some cut-off value and feeds the tables into the R package mada. There are two problems:

(2a) The mada package indeed implements the original Reitsma et al. (2005) approach. The original model is a generalized linear mixed model (GLMM) which was hard to fit in 2005 and hence Reitsma et al. have approximated their own model with an LMM (=linear mixed model), inspiring the implementation in mada. Almost all other current implementations (e.g. in Stata metandi and in some other R packages) use a GLMM. The difference is unfortunately non-negligible, as a simulation study of Vogelsang et al. (2018) reports. This will create a small bias in the analyses and is hence a limitation. From what I know, there is no current implementation in R, though the outdated package Metatron did provide that for some time (<https://cran.r-project.org/web/packages/Metatron/index.html>).

(2b) More serious than (2a) is the following conceptual issue: You seem to have access to raw data from the other primary studies. This means, the whole ROC-curve is available. You reduce the curve to a single point (this is what the Reitsma Model can handle). There is a loss of information (e.g., using three or more points gives some insight into ROC shape and leads to more stable estimates). The diagmeta package by Steinhäuser et al. (2016) is one current approach, but it has the drawback that cut-off values have to be assumed to be on the same scale (which might be implausible given lab procedure differences). Another approach by Doeblner & Holling (2014) reduces the study-level ROC-curves to an accuracy and a shape parameter and has recently been used in a meta-analysis of microRNAs (Sehovic et al., 2022). Hoyer et al. (2021; <https://doi.org/10.1002/bimj.202000091>) compare some more models. I suggest to employ one of these strategies or a comparable one to increase the precision of your findings.

Minor Comments

#####

3) There are problems with some figures where points overlap. Consider a mild jittering to avoid that.

4) The network analysis not very conservative. It produces lots of edges and I am not convinced that we see much structure.

5) It is a great idea to provide a github link. However, I could not access github, since I could not validate my device (someone got send a code to their mobile).

6) I like that you provide a clean Shiny App. It lacks some pointers to the methods and my second concern is that it's long-term access seems to hinge on a single person hosting it. Is there a way to run it on some kind of (open access) platform?

References

=====

Doebler, P., & Holling, H. (2015). Meta-analysis of diagnostic accuracy and ROC curves with covariate adjusted semiparametric mixtures. *Psychometrika*, 80(4), 1084-1104.

Sehovic, E., Urru, S., Chiorino, G. et al. Meta-analysis of diagnostic cell-free circulating microRNAs for breast cancer detection. *BMC Cancer* 22, 634 (2022). <https://doi.org/10.1186/s12885-022-09698-8>

Steinhauser, S., Schumacher, M. & Rücker, G. Modelling multiple thresholds in meta-analysis of diagnostic test accuracy studies. *BMC Med Res Methodol* 16, 97 (2016). <https://doi.org/10.1186/s12874-016-0196-1>

Vogelgesang F, Schlattmann P, Dewey M. The Evaluation of Bivariate Mixed Models in Meta-analyses of Diagnostic Accuracy Studies with SAS, Stata and R. *Methods Inf Med*. 2018 May;57(3):111-119. doi: 10.3414/ME17-01-0021. Epub 2018 May 2. PMID: 29719917.

Response to Reviewers' comments:

Reviewer #1 (Remarks to the Author):

This investigation by Babačić et al describes a mass-spectrometry based analysis of COVID-19 patient serum, in which they attempt to determine altered biomarkers associated with disease severity. These biomarkers were then compared to findings in an *in vitro* model of SARS-CoV-2 infection to confirm any alterations were infection based. They also describe a meta-analysis, in which the authors have compared 11 mass spectrometry-based databases to compare findings and identify commonly occurring biomarkers. This is a well written manuscript and describes an interesting methodology for maximizing the number of identifiable proteins detectable in serum. Whilst, the manuscript describes an in-depth approach to characterising the Covid-19 infected proteome the findings are not novel as many other studies that have done this and are quoted and referenced by the authors themselves. It would be better to see a further application of the methods that can give a unique biological insight about covid19 infection such as comparison with other infections, effect of prior vaccination or prediction of long-term symptoms.

In addition, the experimental design is lacking sample numbers for robust statistical analysis. It is appreciated the profiling method is long and intensive and the authors have tried to address this by validating using cell work and meta-analysis of other studies, but these only confirm what is already described thereby lacking in novel findings. It would of been better to see any novel significant proteins/pathways identified of interest confirmed on much larger sample numbers using a targeted method such as ELISA or MRM-LC-MS/MS. This manuscript in its current form would be better revised to be a methodological study in another journal. For these reasons I do not recommend this manuscript suitable for publication in Nature communications.

We thank the reviewer for the feedback and appreciation of the method and writing. Whereas mass-spectrometry or affinity-based proteomics studies of the plasma/serum proteome have been performed before, in this manuscript we demonstrate several novel methodological and biological findings. To better show the uniqueness of our study we have revised the manuscript, included additional experiments, datasets, and analyses, and rewritten the text to better highlight the novel findings.

1. Firstly, we demonstrate that our method has the largest coverage of MS methods (Figure 5A). This depth of the coverage itself is particularly relevant because it allowed us to find novel alterations that have not been described before, such as the increase in proteasomal proteins (such as PSMA7, PSME1, PSMB3, PSMB5, and PSMB8). The proteasomal proteins are involved in antigen presentation and are associated with interferon responses, which has been implicated as one of the underlying mechanisms behind severity in COVID-19. We also show that higher levels of proteasomal proteins correlate to higher immune response while at the same time correlating to clinical markers of severity. We have further highlighted these findings in the discussion, to help guiding the reader why proteasomal proteins are a relevant novel finding (page 16, line 494).
2. Secondly, we present novel insight into the potential source of the altered circulating proteins by connecting the serum analysis to an *in vitro* experiment,

using SARS-CoV-2 infected cells. As we have referenced in the manuscript, a previous attempt at in-depth MS analysis of the cellular proteome alterations induced by SARS-CoV-2 did not detect alterations in interferon-induced and proteasomal proteins, possibly due to the short follow-up. By adding additional, later, time points we pick up alterations that have not been described before. In the revised version of the manuscript, we have further expanded this tracing of serum proteome alterations to proteome alterations occurring in organs of COVID-19 patients, where we find the proteins downregulated in the white pulp of the spleen to have an increase in the blood of COVID-19 patients. Please refer to more details in our response to the questions raised by Reviewer 2.

3. To further show the usefulness and versatility of our in-depth mass spectrometry-based approach, in the revised version of the manuscript we have explored the serum data for phosphorylated proteins and included a phosphoproteomic experiment of the SARS-CoV-2 infected cell lines in parallel. In this experiment we show for the first time phosphosites on serum proteins implicated in COVID-19 pathophysiology, specifically phosphorylation of calnexin, a chaperone that like the proteasomal proteins is involved in regulating protein degradation and antigen presentation. We report different dynamics in the levels of phosphorylated calnexin compared to the non-phosphorylated protein in COVID-19, which has not been described before.
4. Finally, we acknowledge that it is common in MS proteomics that researchers would perform global proteomics at some depth, select several targets with the strongest signal, and then perform targeted proteomics on a larger sample with ELISA or MRM LC-MS/MS or PRM LC-MS/MS. In this manuscript, instead of selecting 5 or 10 targets that are most likely to succeed in validation and introducing selection bias, our strategy has been to increase the number of study participants by performing an extensive meta-analysis, analysing all the proteins that have been identified in at least two global proteomics studies, in a relatively large sample size of 1,706 individuals from different ethnic backgrounds and different countries. These findings have a higher degree of external validity than a monocentric proteomics study. In this manner, we address the limitation of the low number of study participants in our proteomics analysis and show that the findings that are overlapping with HiRIEF are of high confidence, including not just in estimating the direction of the change but also in the effect size of the change – please refer to Figure 6C, where the log₂-FC by HiRIEF had high correlation (Spearman's $r = 0.82$) with the meta-analysis SMD estimate.

In the new manuscript version, we have expanded the analysis by including more datasets and further highlighted and clarified that the validation of our findings was not the only aim in performing the meta-analysis. With the meta-analysis we aimed for a more comprehensive and conclusive analysis of plasma proteins in COVID-19, which clarifies conflicting results and finds significant alterations where the underlying studies have been underpowered to detect a signal, as exemplified with LBP, B2M, PSMB5, PSMB8, and IGHV3-23. These discoveries demonstrate how the meta-analysis can detect alterations in proteins that are very important components of immune response, thus unravelling the pathophysiology of the disease.

The meta-analytical approach itself is a novelty in global plasma/serum proteomics that can aid the COVID-19 and proteomics research community. It provides concluding estimates for 1,517 proteins, answering the biological questions: “Summarising the current knowledge, is this protein altered in COVID-19, how much, and how useful it could be in differentiating COVID-19 disease status?”.

To make the meta-analysis useful for the scientific community we have created the CoViMAPP resource to present results from a meta-analysis and provide summarising estimates that would be valuable to anyone who is working on circulating proteins in COVID-19, and potentially in other infections.

We believe that these reasons highlight why a multifaceted manuscript that provides both novel biological insights, methodological improvements, validation of its own and other findings, and connecting multiple sources of information would benefit the wide readership of a high-quality multidisciplinary journal that *Nature Communications* is.

Minor concerns

1. For the meta-analysis comparing serum and plasma proteomic datasets – is this appropriate for two distinct biofluids? Sensitivity can be much greater in serum due to lower amounts of high abundant proteins in plasma, coupled with the Top 14 depletion that has been carried out in the serum analysis in this investigation. This should be clarified.

This is a very interesting question. Serum is often referred to as blood plasma without clotting factors. This makes serum a component of plasma. The major differences in the protein identification sensitivity in the separate MS studies were most likely due to methodological approaches.

However, the common definition of serum is also debatable. As one can see in CoViMAPP, many of the clotting factors are not only detected in several serum proteomics studies, but their direction of alteration is the same as observed in plasma – we would specifically pinpoint VWF and F13A, as the strongest signal. Thus, maybe a better definition of serum would be plasma depleted of clotting factors, where the systematic effect of the depletion in clotting proteins by coagulation does not diminish the quantitative differences between disease and control.

Since the meta-analysis summarises the effect sizes of the alterations in separate studies, i.e., the mean differences in comparisons between COVID-19 patients and PCR-negative controls analysed by the same MS method in the same type of fluid, it is valid to include both serum and plasma studies, thereby increasing the power of the analysis and the coverage of proteins. We have previously provided a modularity within CoViMAPP where the user can specifically test differences between serum and plasma studies. We have further performed stratified analyses based on sample type (plasma or serum) and found no major differences. These analyses are now included as supplementary figures in Figure S17.

2. The authors seem to use serum and plasma proteome interchangeably. They have analysed serum proteome themselves but refer to changes in plasma proteome.

Please refer to our previous answer. We have used the term serum proteome when we report results found in our study and plasma proteome when referring to the meta-analysis results. Since serum is a component of the plasma, we deemed the term plasma proteome more accurate for the meta-analysis. However, we agree that this might be confusing to the reader; thus, in the revised manuscript we introduced the word “soluble blood proteome” that would be a more accurate description of the samples.

3. Methods, Solvent A detailed several times as 5% water, 95% acetonitrile, 0.1% formic acid, totalling 100.1%. Solvent B detailed several times as 5% water, 5% DMSO, 95% acetonitrile, 0.1% formic acid totalling 105.1%.

Thank you for spotting this error. It should state Solvent A: 94.9% water, 0.1% formic acid, and Solvent B: 5% acetonitrile, and Solvent B: 4.9% water, 0.1% formic acid, 5% DMSO, and 90% acetonitrile. This has been updated in the revised version of the manuscript.

4. Mass spectrometer conditions need to be more detailed (source temp etc)

We have now added source temperature.

5. Intermittently use uL instead of μ L throughout the methods section

We apologise for this error. We have now corrected it.

Reviewer #2 (Remarks to the Author):

The manuscript analyzed the proteome of serum samples collected from 20 hospitalised patients with COVID-19 infected with the ancestral SARS-CoV-2 variant, and 7 healthy controls, which were PCR-negative and seronegative, using HiRIEF-based extensive fractionation and TMT-based quantitative proteomics. By performing a deep fractionated (72 fractions per batch, LC gradient unknown) TMT-based proteomic profiling, the author identified for more than 2000 proteins, which is deeper compared with most other LC-MS-based proteomics studies. Then they performed bioinformatics analysis of the proteomics data and tried to analyze the tissue origin of plasma proteins, but these bioinformatic analyses are descriptive and hypothetical. So does the comparison with PEA and SOMAscan data sets. Then the authors performed an in vitro cell line proteomic study with a deep proteomic coverage and claimed this could be used to gain insights into which proteins derived from SARS-CoV-2 infection site. This does not make much sense to the reviewer, because cellular proteome of cell line data can be hardly associated with circulating proteins in COVID-19 patients. With a FDR <5%, the authors performed bioinformatic analysis and discussed some of the regulated proteins when the cell line was perturbed by SARS-CoV-2. Finally, the authors claimed via meta-analysis with existing proteomic data sets, they explored the potential of selected proteins for COVID-19 diagnosis.

We thank the reviewer for the feedback. Tracing plasma proteins to their tissue origin is perhaps as challenging as the attempt to delineate proteome changes in tissues from contamination by plasma or the blood in the case of an infectious disease such as COVID-19. Thus, we have further clarified in the text that the aim of the infection experiment was to provide a “clean” system, isolated from the systemic effects in COVID-19, thus providing higher confidence in tracing the serum alterations to SARS-CoV-2 infection. In the revised manuscript, we have added two published tissue proteomics datasets as comparison to both the serum proteome and cellular proteome alterations, demonstrating that a set of proteins can be traced to a SARS-CoV-2 infection at different levels – the cellular, tissue, and systemic in blood (Schweizer *et al.* 2022, medRxiv; Nie *et al.* 2021, Cell). We have performed this analysis after removing any protein annotated as plasma or immunoglobulin protein by Schweizer *et al.* (2022). In addition, bioinformatically, we have described which of the altered serum proteins are more likely to derive from a specific tissue based on the analyses of the Human Protein Atlas, where proteins are annotated as tissue-enriched if they had a four-fold increase in mRNA levels compared to any other tissue in the body. We have performed this analysis because the mRNA data provide orthogonal information which tissues produce which proteins.

Major issues:

1, In general, this is a comprehensive proteomics study in terms of proteomic depth and bioinformatic analysis. However, the main contribution of this paper to the field is unclear. The patient sample size is relatively small, and the clinical value is minimal. The vague term “diagnosis” in the last section does not make much sense to the reviewer, because it is not a clinical need to diagnose COVID-19 by blood proteins. If the authors are to show that a protein-based model outperforms existing methods for diagnosing COVID-19 in the clinic, more experimental data are required. Differentiation between

severe and non-severe COVID-19 cases is a clinical need, but multiple papers have been published in the field. This manuscript does not show its uniqueness in this regard neither.

We argue that the contribution of this manuscript to the field lies in the depth of the analysis, the tracing of the origin of plasma proteins, the statistical integration of the different studies into a meta-analysis and providing a resource for the research community. The added value in the new version of the manuscript lies in the most up-to-date meta-analytical summary of plasma protein alterations, in further tracing the signal to tissues of COVID-19 patients, and phosphoproteomics.

We agree that the direct clinical application of the manuscript is limited. However, this was a discovery omics study that describes changes in the systemic biological processes and aimed to identify which alterations in the serum comes from the proteome alterations induced by SARS-CoV-2 infection, both to understand the changes triggered by disease and to better understand the plasma proteome in a case study of infection. We present several novel findings on the alterations such as the increase in proteasomal proteins, tissue-enriched proteins, specifically proteins derived from the white matter of the spleen, and decreased level of phosphorylated calnexin in the serum of COVID-19 patients.

The value of the meta-analysis is not only in addressing the limitation of the small sample size of our cohort but also in the ability to detect alterations not picked up by the underlying, potentially underpowered studies. Furthermore, it has a higher degree of external validity and the approach itself is a novelty. The CoViMAPP resource will provide researchers with information on whether a protein of their interest is alternating in COVID-19, how much, and how helpful it is in differentiating COVID-19 disease status.

Furthermore, the SROC meta-analysis provides higher level of evidence about how common diagnostic protein biomarkers, such as CRP, LDH, and ferritin, might perform compared to other circulating biomarkers. As we point out in the manuscript, some proteins, such as CRTAC1, SERPINA3, ORM1, and TF, might perform better as non-specific circulating protein biomarkers of COVID-19, and possibly other infections, than those currently used. This showcases how unbiased analyses of the systemic pathophysiology behind the disease by analysing the plasma proteome can pinpoint new targets for future research and repurposing biomarkers for clinical use.

We agree with the reviewer that using the word diagnostic biomarkers might confuse the reader that it implies a validated diagnostic biomarker. Therefore, we have rewritten the discussion, to avoid this confusion.

The clinical need for comparing patients of different severity is relevant, however this is outside of the scope of the manuscript. Although the COMBAT consortium has performed a comprehensive multiomics analysis of high quality to address this question (2022, Cell), it would be very interesting to address it in a meta-analysis. However, this carries a major challenge of delineating the effects of severity by harmonising different groups of severity analysed in different studies and different hospital settings.

To further show the usefulness and versatility of our in-depth mass spectrometry-based approach, in the revised version of the manuscript we have explored the serum data for

phosphorylated proteins and included a phosphoproteomic experiment of the SARS-Cov-2 infected cell lines in parallel. In this experiment we show for the first time phosphosites on serum proteins implicated in COVID-19 pathophysiology, specifically phosphorylation of calnexin, a chaperone that like the proteasomal proteins is involved in regulating protein degradation and antigen presentation. We found different dynamics in the levels of phosphorylated calnexin as compared to the non-phosphorylated protein in COVID-19, which has not been described before

2, The virus-infected cell line proteomic experiments should be applauded, but its association to the plasma proteome data set seems weak, if not far-fetched. The meta-analysis is useful but did not bring much new insights nor clinical value.

We thank the reviewer for their feedback and have re-written the manuscript to better explain the rationale behind the importance of deconvoluting the systemic plasma proteome data.

We argue that it is important to differentiate which alterations in blood proteins are leakage deriving from infection and which from secondary processes triggered by infection, e.g., proteome alterations in immune cells or proteome alterations due to tissue damage caused by the virus. We demonstrated this through an experiment by proteomic analysis of infected cells *in vitro* because proteomic analysis of tissue samples obtained from infected patients raises several sources of variability, making it difficult to claim that the proteome alterations derived from infection. Schweizer *et al.* (2022, medRxiv) have attempted to address this question in another study and faced these challenges, showing that the circulating proteins are an integrated part of the tissue proteome. In the revised version of our manuscript, we use datasets derived from their study and another (Nie *et al.*, 2021, Cell) to trace the altered proteins to tissues of COVID-19 patients. Some of the strongest signals in the *in vitro* experiment and the plasma proteomics analysis are traceable to different tissues, and we find an interesting relation between proteome alterations in the white matter of the spleen, where the protein loss might be explained by their release in the blood during the host's immune response to SARS-CoV-2 infection.

The analyses in the first submission were based on a logical hypothesis – changes in protein levels in the blood that derived from infection itself should occur in the same direction as the changes in protein levels in SARS-CoV-2 infected cells. In this version, we show that many of the alterations in the infection experiment are traceable to changes in specific organs at protein level. However, by using an isolated system we can be more confident that the changes are due to infection and not blood contamination or effects of inflammation.

We thank the reviewer for appreciating that the meta-analysis is useful, such as validating our findings and being an important resource for the community. However, we disagree that the meta-analysis did not add much new insights. First, on several instances including LBP, HSP90A1A and PSMB8, it clarifies conflicting results reported in different studies. Second, it picks up signals where most studies have shown statistically non-significant findings due to being underpowered (e.g., B2M, IGHV3-23, PSMB5). Third, it shows that the findings are not serum or plasma specific. In addition, we have now updated the meta-analysis to a search that dates to end of February 2023 and included ten additional datasets. This has further improved the meta-analytical estimates and unraveled additional novel

findings, providing clarifications in the alterations in the blood levels for many proteins, as mentioned above.

Minor issues:

1. More technical details for the procedures for tissue-enrichment analysis should be provided. Why only 1779 out of 2037 proteins were used for HPA enrichment analysis?

We used only these proteins because we have statistically tested only 1779 proteins that had observations in at least 50% of the individuals.

2, The statement of first meta-analysis of COVID-19 plasma proteome as well as its comprehensiveness might be over-claimed. Though described in the method section and Figure S10, it is not clear how the studies were collected and how the datasets were systematically normalized and curated. Please add more details so that this section could be potentially reproduced.

We described the methodology behind the systematic review in Figure S10 (now Figure S16) and the methods section, along with specific inclusion and exclusion criteria in the curation process and the numbers excluded at specific stages:

“On 24-02-2022, we searched for studies of interest in two databases, i.e., PUBMED and EMBASE, for the keywords: “(covid-19 OR sars-cov-2) AND (plasma OR serum) AND mass spectrometry AND (proteomics OR protein*)”, without restriction. The references were handled in Mendeley reference manager. After excluding the duplicates, we screened the articles manually based on inclusion and exclusion criteria. The inclusion criteria were studies with a global MS method on plasma or serum of human COVID-19 patients and PCR-negative human controls. We excluded reviews and conference articles. Studies selected in the screening step were then further evaluated for inclusion in the review and meta-analysis based on reading the full papers and accessing the publicly available processed normalised data from proteomic searches. Where processed data were not available, we contacted the first and/or corresponding authors at least three times to ask for access to the data. If the authors did not respond or did not provide the data, the corresponding study was not included in the meta-analysis. Studies that have pooled samples before the MS analysis were excluded from the meta-analysis because the calculation of standard deviation would not have been valid.”

We have now performed a new systematic review and updated the meta-analysis with new datasets published as recent as beginning of 2023 and added further information to the methods section. For each dataset we have now provided exact information on how the dataset was obtained and how we processed the data for each study. The processing file is now available for download from CoViMAPP and github.

3, Have the authors provided the LC-gradient length for the LC-MS analysis? Not clear in the sections “LC-ESI-MS/MS Q-Exactive HF” and “Mass-spectrometry cell proteomics”.

We apologise for not providing this information in the previous version. We have now clarified this in the methods section and supplementary information, Table S23.

4, Please provide more details of the criteria for inclusion and exclusion of a patient samples for this specific study, especially for healthy controls.

We have now provided information on the inclusion and exclusion criteria in the methods section.

5, Please provide more details for the shiny app and provide a user manual.

We provided the code for the shiny app and description on how to interpret and use each parameter in the shiny app. We have now added more details on how to use and interpret the findings in the shiny app and provided a manual.

6, “Automatic gain control (AGC) targets were 1×10^6 for MS1 and 1×10^5 for MS2.” - -- should be 10^{*6} and 10^{*5} (superscript).

Thank you for spotting this error. We have now corrected it.

7, Subtitle: “Mass-spectrometry cell proteomics”: may be changed to Mass spectrometry-based proteomics analysis of cells

Thank you for the suggestion. We have now changed it.

Reviewer #3 (Remarks to the Author):

Review of "Comprehensive proteomics and meta-analysis of COVID-19 host response detects elevated proteasomal proteins in blood traceable to SARS-CoV-2 infection"

I am not an expert on proteins (not even close). I have been asked to contribute a review of the statistical methodology and since I am not working in bioinformatics, the focus of the review will be on the meta-analytical methodology.

The manuscript studies proteins in persons with and without a CoViD infection using mass spectrometry. It has three main parts: (i) An original study with relatively small sample size but a large number of proteins, (ii) an in vitro experiments with infected cells, and (iii) an elaborate meta-analysis that adds data from other mass spectrometry studies. Given my own specialization, I have only two comments on (i) and (ii) but I can provide detailed feedback on (iii). Overall, the meta-analytical strategy of providing analyses for such a large number of biomarkers is laudable, but as implemented, it fails to leverage the complete potential of the data (see below).

We thank the reviewer for the very constructive feedback and the appreciation of the extent of the meta-analysis.

Major Comments

=====

1) In your original study and elsewhere you use the customary t-tests and an FDR of 5% or 10% to screen for relevant proteins. The problem with the t-test, especially in the small sample size situation with 20 CoViD and 7 controls, is that we can never really check any distributional assumptions. Asymptotic normality is questionable, too. There might be proteins with small absolute mean differences but due to underlying skewness or random chance, the sample variance is low in both groups. This in turn reduces the denominator in the t-statistic, inflating p-values. I recommend contrasting a more conservative procedure, e.g., the empirical Bayes approach in the LIMMA package (not sure if it works off-the-shelf though for MS data). I understand that this creates some work downstream, but since meta-analyses like these have the potential to point others in the wrong direction, a somewhat more conservative screening is advisable given the potential impact.

We agree with the reviewer that it is difficult to work with small sample sizes when it comes to the underlying assumptions. In the initial version of the manuscript, we have used a t-test for univariate comparisons and limma for multivariate analyses between COVID-19 and controls, where we adjusted for age, sex, and comorbidities.

We decided for a univariate t-test in both the serum and cell-line comparisons because the results showed that it was more conservative.

For the comparisons in serum, we identified 619 differentially altered proteins (DAPs) with a t test, and 715 DAPs with univariate limma. The overlap between the tests was 561 proteins. Assuming the proteins that don't overlap are false discoveries, would mean that the t test led to 58 additional hits (9.37%) by inflating p values, as compared to limma, which had 154 additional hits (21.54%).

For the comparisons in the cell lines, the difference was much larger. For the proteome alterations 3 days post infection, 521 proteins were altered based on a t test, 1,868 based on limma, and 1,862 based on DEqMS (please refer to the Venn diagrams in Figure R1 below). DEqMS is a statistical method based on the limma approach but adapted for MS data, where it uses the number of peptide spectra matches (PSMs), to adjust the variance. This time, based on the assumption that proteins that do not overlap with at least one other method are false discoveries, 50 proteins were DAPs based only on a t test (9.60%), 1074 proteins were DAPs based only on limma (57.49%), and 1389 proteins were DAPs based only on DEqMS (74.60%). Of the 42 proteins overlapping with the serum t test analysis, only one was identified solely by a t test of the cellular proteomics comparison. Compared to the other methods - 63 out of the 126 proteins overlapping with DAPs in serum were limma specific (50%), and 64 out of the 94 proteins overlapping with DAPs in serum were DEqMS specific (68.1%).

We observed a similar overlap analysing the proteome alterations at day 7, although without t-test specific DAPs overlapping with the serum analysis (please refer to the Venn diagrams in Figure R2 below).

Altogether, in this data context, based on the results, we were more confident in the signals identified by the t test, which appeared more conservative. This could be driven by a larger measured variance in the data that leads to higher p values as estimated by a t test. One possible explanation for the much larger number of hits based on the other statistical methods could be that the variance smoothing in limma and DEqMS inflates the p values, making mean differences of smaller effect sizes statistically significant. We did not perform DEqMS in serum because it would penalise the lower abundant proteins, which are more likely to have lower count of PSMs but are of greater biological interest to explore in the serum.

Figure R1

Figure R2

2) The diagnostic meta-analysis you perform takes the raw data, builds a fourfold table from that at some cut-off value and feeds the tables into the R package mada. There are two problems:

(2a) The mada package indeed implements the original Reitsma et al. (2005) approach. The original model is a generalized linear mixed model (GLMM) which was hard to fit in 2005 and hence Reitsma et al. have approximated their own model with an LMM (=linear mixed model), inspiring the implementation in mada. Almost all other current implementations (e.g. in Stata metandi and in some other R packages) use a GLMM. The difference is unfortunately non-negligible, as a simulation study of Vogelsang et al. (2018) reports. This will create a small bias in the analyses and is hence a limitation. From what I know, there is no current implementation in R, though the outdated package Metatron did provide that for some time (<https://cran.r-project.org/web/packages/Metatron/index.html>).

We are very grateful for this comment. We have now implemented a GLMM-based model, using the glmer package and following the approach by Sehovic *et al.* (BMC Cancer, 2022). This has, on average, improved the estimates, giving higher values of sensitivity and specificity obtained in the bivariate model (please refer to Figure R3 below).

Figure R3

(2b) More serious than (2a) is the following conceptual issue: You seem to have access to raw data from the other primary studies. This means, the whole ROC-curve is available. You reduce the curve to a single point (this is what the Reitsma Model can handle). There is a loss of information (e.g., using three or more points gives some insight into ROC shape and leads to more stable estimates). The diagmeta package by Steinhäuser et al. (2016) is one current approach, but it has the drawback that cut-off values have to be assumed to be on the same scale (which might be implausible given lab procedure differences). Another approach by Doeblner & Holling (2014) reduces the study-level ROC-curves to an accuracy and a shape parameter and has recently been used in a meta-

analysis of microRNAs (Sehovic et al., 2022). Hoyer et al. (2021; <https://doi.org/10.1002/bimj.202000091>) compare some more models. I suggest to employ one of these strategies or a comparable one to increase the precision of your findings.

We agree with the reviewer that this is an issue in SROC curves meta-analysis and that the Steinhauser *et al.* (2016) approach is not valid in this instance. In this version, we have implemented the Doebler & Holling (2015) approach of estimating the α and C1 parameters, as used in the study by Sehovic et al. (2022). Because we had the whole ROC curve, we performed a minor modification by adding more pairs of sensitivity and specificity on a study-specific ROC curve to estimate the alpha and C1 parameters (corresponding to specific cut-off points). Instead of 3, we used $n = m + 1$ pairs of sensitivity and specificity, where m was the number of observations for the given protein in a specific study that were not missing values. We have now appended the values of C1 and α for minimal heterogeneity Q in the summary table in CoViMAP for each study and each protein, as estimated by the Doebler & Holling (2015) approach. The averages of the per-protein α estimates ranged from 0.49 to 1.38, with median and mean value of 0.93 (IQR: 0.84-1.00).

We based our interpretation on the α estimates because they appeared more stable. We observed a minor effect of α estimates of the underlying ROC curves in the meta-analytical estimates of sensitivity and specificity (Figure 6F). Therefore, we kept the value of 1 for alphasens and alphafpr in plotting the SROC curves.

Minor Comments

#####

3) There are problems with some figures where points overlap. Consider a mild jittering to avoid that.

We have now introduced jittering where possible.

4) The network analysis not very conservative. It produces lots of edges and I am not convinced that we see much structure.

There was a large co-correlation between the proteins, which led to a lot of edges between them. The edges are based on the Spearman correlation coefficient of $r > 0.8$. We have now removed the network analysis and replaced it with a gene-set enrichment analysis of gene sets defined by their alterations in the infected cell lines (Figure S10C).

5) It is a great idea to provide a github link. However, I could not access github, since I could not validate my device (someone got send a code to their mobile).

We sincerely apologise for this inconvenience. The github access should be available now and we are attaching the code as a zip file for review to this submission as well.

6) I like that you provide a clean Shiny App. It lacks some pointers to the methods and my second concern is that it's long-term access seems to hinge on a single person hosting it. Is there a way to run it on some kind of (open access) platform?

This is a very good point and we have now arranged to meet this requirement. Upon publication, the shiny app, datasets, and code will be available through the Scilifelab data centre platform (<https://www.scilifelab.se/data/>). This will provide a long-term access to CoViMAP. During the review process the app is still available only on proteomics.se,

but once accepted for publication, the entire app will be transferred and publicly available at dockerhub (<https://hub.docker.com/>) and made available via Scilifelab, without password protection. To avoid issues with the different links during publication and transferring the app, we have created a doi link for the app (<https://doi.org/10.17044/scilifelab>).

REVIEWERS' COMMENTS

Reviewer #1 (Remarks to the Author):

The authors have made great efforts highlight new findings in their in depth proteomics analysis with new proteins and inclusion of a phosphoproteomics analysis which is interesting. However my original concern on the lack of statistical power still stands.

The authors propose that the meta-analysis is adequate to validate their findings and its fortunate that there have been many other studies conducted for the authors to do this. However this only validates for changes previously described and means the novel findings the authors have now described in their data set regarding the proteasomal proteins are not validated by any other study therefore only based on n=20 covid samples and n=7 controls. This is where a quicker targeted 'monocentric proteomics' would be valuable to confirm the novel finding the authors now claim.

From a biological perspective serum and plasma are defined by how they are collected and the associated proteomes have always known to be different. It is fine if both types have been used and no difference in DEPs is observed in the changes as shown by the authors but it is good practice to clarify what blood component has been studied especially if the authors claim that a diagnostic blood test could be created from this analysis.

Minor comments

Figure 1. Can authors make it clear of the proteins in 1A are differential proteins detected or all proteins available in the PEA and somlogic panels

Figure 2 D and E please give a legend for the colors on the plot. Were these the same cohort of samples analysed the the 3 technologies?

Figure 4c correct interpheron to interferon

The authors need to provide a better description of figure 5D. Is this a map showing degree of overlap between studies?

Reviewer #2 (Remarks to the Author):

Thanks for the revisions. I still think the sample size is small and the association among the blood proteome, cell line experiments and the meta-analysis is weak. Tracing protein released from a

particular tissue in the blood would require much more data, since the blood contains proteins released from almost every tissue type, and the kidney also modulate protein concentrations in the blood. I suggest to tone down this part. Otherwise, I think the paper presents a useful data resource, and support its publication in Nature Communications.

Reviewer #3 (Remarks to the Author):

The authors have addressed all the issues I raised with great attention to detail. I can now understand, why the more conservative t-test was used.

Especially the supplemental material was helpful to ascertain that this project was cleanly executed. I also had another look at CoviMAPP and felt it was potentially very helpful.

Based on how the authors handled my remarks, I am confident that CoviMAPP is a solid, scientific contribution.

However, I could still not access the github (because of the two factor authentication for the gmail account). I can hence not review the code behind the project.

As I wrote earlier, I cannot really assess the protein details nor the MS details.

REVIEWERS' COMMENTS

Reviewer #1 (Remarks to the Author):

The authors have made great efforts highlight new findings in their in depth proteomics analysis with new proteins and inclusion of a phosphoproteomics analysis which is interesting. However my original concern on the lack of statistical power still stands. The authors propose that the meta-analysis is adequate to validate their findings and its fortunate that there have been many other studies conducted for the authors to do this. However this only validates for changes previously described and means the novel findings the authors have now described in their data set regarding the proteasomal proteins are not validated by any other study therefore only based on n=20 covid samples and n=7 controls. This is where a quicker targeted 'monocentric proteomics' would be valuable to confirm the novel finding the authors now claim.

From a biological perspective serum and plasma are defined by how they are collected and the associated proteomes have always known to be different. It is fine if both types have been used and no difference in DEPs is observed in the changes as shown by the authors but it is good practice to clarify what blood component has been studied especially if the authors claim that a diagnostic blood test could be created from this analysis.

We thank the reviewer for the comments.

We agree with the reviewer that targeted proteomics analyses would be further useful to increase the certainty of the finding and that should be a focus for future studies.

The methodological details and the sample type (serum or plasma) are clarified in CoViMAPP, and we have further performed sensitivity analyses based on this.

Minor comments

Figure 1. Can authors make it clear if the proteins in 1A are differential proteins detected or all proteins available in the PEA and somlogic panels

This has been clarified in Figure 1A (now Figure 2a).

Figure 2 D and E please give a legend for the colors on the plot. Were these the same cohort of samples analysed with the 3 technologies?

Thank you for spotting this. The legend for the colours is now provided for the Figures 2D and 2E (now Figures 3d and 3e). The cohorts analysed by Olink and SomaScan are different between themselves and from our cohort.

Figure 4c correct interferon to interferon

Thank you for spotting this. We have now corrected it.

The authors need to provide a better description of figure 5D. Is this a map showing degree of overlap between studies?

That is correct. We have now clarified this for the figure 5D (now 6d).

Reviewer #2 (Remarks to the Author):

Thanks for the revisions. I still think the sample size is small and the association among the blood proteome, cell line experiments and the meta-analysis is weak. Tracing protein released from a particular tissue in the blood would require much more data, since the blood contains proteins released from almost every tissue type, and the kidney also modulate protein concentrations in the blood. I suggest to tone down this part. Otherwise, I think the paper presents a useful data resource, and support its publication in Nature Communications.

We thank the reviewer for the comments. We agree, and have now tried to further point out that tracing proteins from a particular tissue or infection is rather difficult to do with certainty and that we in this work report findings that suggest that some of these proteins derive from infection. We have added this paragraph in the discussion: "It is worth reminding the reader that although we find similar protein alterations that occur in the serum of COVID-19 patients, after infection in vitro, and in different tissues obtained from deceased COVID-19 patients, these do not necessarily prove with certainty that the alterations are specifically and directly deriving from SARS-CoV-2 infection or a specific organ. However, the evidence supports the hypothesis that some of the observed proteome alterations in the blood of COVID-19 patients are likely derived from SARS-CoV-2 infection, whereas some appear to be shared across different organs."

Reviewer #3 (Remarks to the Author):

The authors have addressed all the issues I raised with great attention to detail. I can now understand, why the more conservative t-test was used.

Especially the supplemental material was helpful to ascertain that this project was cleanly executed. I also had another look at CoviMAPP and felt it was potentially very helpful.

Based on how the authors handled my remarks, I am confident that CoviMAPP is a solid, scientific contribution.

However, I could still not access the github (because of the two factor authentication for the gmail account). I can hence not review the code behind the project.

As I wrote earlier, I cannot really assess the protein details nor the MS details.

We thank the reviewer for the comments. We realised that it was a gmail security issue when we received a notification from Google that the gmail account is blocked due to the log-in attempt, but despite our request to Google, they could not remove the two-factor authentication. The github link is now publicly available.